# Silicon-Oxygen Region Infrared and Raman Analysis of Opals: The Effect of Sample Preparation and Measurement Type

Neville J. Curtis [1,2], Jason R. Gascooke [2,*] and Allan Pring [2,*]

1   South Australian Museum, North Terrace, Adelaide, SA 5000, Australia; Neville.Curtis@flinders.edu.au
2   College of Science and Engineering, Flinders University, Bedford Park, SA 5042, Australia
*   Correspondence: Jason.Gascooke@flinders.edu.au (J.R.G.); Allan.Pring@flinders.edu.au (A.P.)

**Abstract:** An extensive infrared (IR) spectroscopy study using transmission, specular and diffuse reflectance, and attenuated total reflection (ATR) was undertaken to characterise opal-AG, opal-AN (hyalite), opal-CT and opal-C, focussing on the Si-O fingerprint region (200–1600 cm$^{-1}$). We show that IR spectroscopy is a viable alternative to X-ray powder diffraction (XRD) as a primary means of classification of opals even when minor levels of impurities are present. Variable angle specular reflectance spectroscopy shows that the three major IR bands of opal are split into transverse optical (TO) and longitudinal optical (LO) components. Previously observed variability in powder ATR is probably linked to the very high refractive index of opals at infrared wavelengths, rather than heterogeneity or particle size effects. An alternative use of ATR using unpowdered samples provides a potential means of non-destructive delineation of play of colour opals into opal-AG or opal-CT gems. We find that there are no special structural features in the infrared spectrum that differentiate opal from silica glasses. Evidence is presented that suggests silanol environments may be responsible for the structural differences between opal-AG, opal-AN and other forms of opaline silica. Complementary studies with Raman spectroscopy, XRD and scanning electron microscopy (SEM) provide evidence of structural trends within the opal-CT type.

**Keywords:** opal; hyalite; geyserite; menilite; infrared; Raman; X-ray diffraction; scanning electron microscopy; gem; silica

## 1. Introduction

Opals are a fascinating area of research, for, although a fundamental classification system [1] has been used for almost 50 years, detailed knowledge of the structural relationships of the silica species involved remains elusive. This is despite the ready availability of both samples and a plethora of analytical techniques such as Raman, infrared (IR) and nuclear magnetic resonance (NMR) spectroscopies and X-ray diffraction (XRD). XRD of powdered samples allows ready classification of amorphous opal (opal-A, comprising "gel"-like opal-AG and "network"-like opal-AN), paracrystalline opal (opal-CT) and a α-cristobalite-like opal (opal-C). Commercial interest is largely based on the property of play of colour (POC) in precious opal and the appealing colour of semi-precious stones. POC arises if the spheres in opal-AG are orderly arranged and of the appropriate size [2], as shown in scanning electron microscopy (SEM) images for the untreated samples presented in Figure 1. Some POC opal is opal-CT, but the nature of the microstructure has not been explored at the same level of detail as opal-AG. Etching of opal-AG (e.g., [2–5]) removes interstitial material to better reveal the spherical arrangements but is not required to determine the arrangement, uniformity and size of the spheres. Despite the dissimilarity of the appearance of opal-AN (hyalite [6]), which has little obvious structure (Figure 1d), when compared to opal-AG (Figure 1a,b), XRD, IR, Raman and NMR features are largely identical, although small differences have been reported in NMR and Raman spectra [7]. The broad and largely featureless XRD patterns of opal-AG and opal-AN imply that the

silica species are amorphous. The nature of opal-CT has been in debate for many years [8,9], with many researchers claiming the structured XRD patterns are a result of a combination of cristobalite- and tridymite-like forms of silica. In previous work [7], we showed that opal-CT exhibits a progressive change from a broader to a sharper XRD pattern structure coupled with more defined Raman spectra with delineated peaks. Examples with a broader XRD pattern may be transparent and exhibit POC. Opal-C shows a XRD pattern and Raman spectrum concordant with α-cristobalite [1], though actual equivalence is tenuous as higher temperature conversion to the β form is not seen [10].

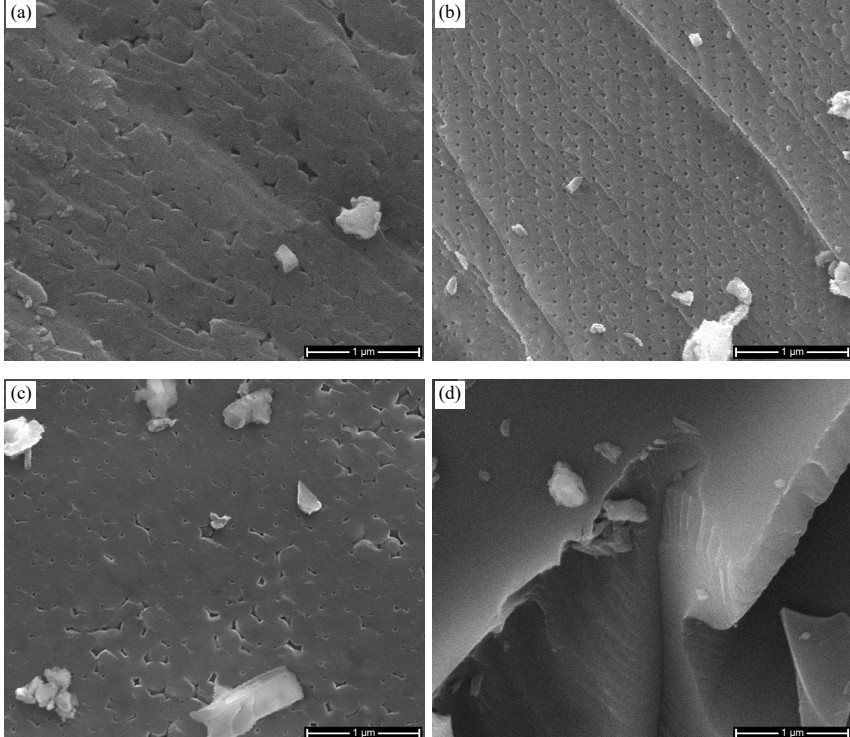

**Figure 1.** SEM images of unetched opal-AG and opal-AN (**a**,**b**) G13767 (collection of opal-AG samples from Lightning Ridge, New South Wales, Australia) showing (**a**) irregular pattern of an opaque blue sample—no POC; (**b**) regular pattern on an orange glass sample but with spheres too small to exhibit POC); (**c**) G8608 (opal-AG, white opal showing POC from White Cliffs, New South Wales, Australia); and (**d**) G32740 (opal-AN (hyalite) from Valec, Czechia).

In a recent spectroscopic and XRD examination of opal [7], we demonstrated that attenuated total reflectance (ATR) IR spectra of powdered samples was an alternative method for distinguishing between opal-A, opal-CT and opal-C. Opal-A has a peak or shoulder at 550 cm$^{-1}$ while opal-C has a small peak at 620 cm$^{-1}$ (see below). Opal-CT can be identified by default as it contains neither of these features.

Three major silicon-oxygen fingerprint bands may be identified that are common to all types of opal and are similar to those found for silica and tetrahedral glasses [11–18]:

- A low-energy band at approximately 470 cm$^{-1}$: ascribed to rocking of the Si-O-Si bridge.
- A medium-energy band at approximately 790 cm$^{-1}$: ascribed to a motion where the bridging oxygen in Si-O-Si moves perpendicular to the line joining the silicon atoms. This has been described as both a bending motion and a symmetric Si-O-Si stretch.
- A higher-energy band at approximately 1100 cm$^{-1}$: ascribed to an antisymmetric Si-O-Si stretch.

In our previous work, we found that there was considerable variability in the spectra, particularly with the 470 and 1100 cm$^{-1}$ peaks for powder ATR measurements. For example, the four samples of opal-A in Figure 1 gave different far-IR spectra in our initial survey even though XRD patterns were identical (see Figure 2 in [7]). This could not be

correlated with other analytic techniques. Similarly, confocal Raman spectroscopy, which also measures surface features, gave no variability of the main peaks between 200 and 500 cm$^{-1}$.

This led us to investigate a set of possibilities for this variability. Our thoughts could be grouped into five categories: particle size effects, intergrowth structures (e.g., layers of cristobalite- and tridymite-like silica [19]), other inherent heterogeneity (e.g., zones of high or low water content), chemical modification introduced through the sample preparation process or a spectroscopic feature peculiar to ATR of powdered samples.

We have thus undertaken an extensive IR-based analysis of opals (see [7,9,16,20–29] for previous studies) to investigate the potential causes of the variability in powder ATR measurements and also to determine whether this can provide insight into the structural nature of the silica present in opals. Specifically, we concentrate on the Si-O region (200–1600 cm$^{-1}$) rather than the water/silanol bands at higher wavenumbers. Thus, we have expanded our initial ATR study under more controlled conditions and incorporated transmission and reflectance measurements.

The form of this paper is to present the raw spectra from the various measurement techniques and then discuss the applicability of IR an as interpretative tool for structural determination and as a standard classification method. Because of the relative scarcity of opal-C specimens, most attention will be focussed on opal-AG, opal-AN and opal-CT. Infrared measurements are complemented by associated XRD, Raman and SEM analysis. We also take the opportunity to provide extensive spectroscopic data to assist in structural interpretation of opal which until now has been mostly concentrated on Raman (e.g., [30]).

## 2. Materials and Methods

Samples were sourced as discussed previously [7] and consisted of museum samples and recently acquired specimens from the more recent exploited deposits. As before, most samples were opal-AG, opal-AN and opal-CT, with very few examples of opal-C available. A limited number of the museum samples had been prepared as polished slides. These included thin (1 to 2 mm) or thick (up to 10 mm) sections mounted on microscope slides or as pieces embedded in resin, cut and polished. Some gem opals were available as facetted or cabochon stones. A variety of IR instrumentation was used. The requirements of the various spectroscopic techniques, such as quantity, size and geometry or presence of a large flat surface, dictate which samples could be analysed. Locality data is given in Table 1 for samples mentioned in text, tables and figures. The results from many more samples are presented in composite terms. Unless otherwise noted, all instruments used in this study were located at Flinders University.

Transmission mode spectra (1 wt% in CsI) were recorded in the far-IR (50–600 cm$^{-1}$) region at the Australian Synchroton's Far-IR beamline using a Bruker IFS 125/HR spectrometer (Bruker, Billerica, MA, USA) at 4 cm$^{-1}$ resolution, and in the mid-IR (400–4000 cm$^{-1}$) region using a Nicolet Nexus 8700 spectrometer (Thermo Electron Corporation, Madison, WI, USA) at 2 cm$^{-1}$ resolution. Spectra were recorded under vacuum for the far-IR measurements and under atmosphere for mid-IR measurements. Peak positions in the overlap region were consistent between the two instruments. At least five different samples were examined for opal-AN, opal-AG, opal-C and over 30 for opal-CT. Additional mid-IR (500–4000 cm$^{-1}$) spectra (1 wt% in KBr) were measured on a Perkin–Elmer Frontier instrument (Perkin-Elmer, Waltham, MA, USA) at 2 cm$^{-1}$ resolution.

Specular reflectance spectra (far- and mid-IR measured consecutively without remounting the sample) were recorded on a Bruker Vertex v80 instrument (Bruker, Billerica, MA, USA). using the 1513/QA attachment under vacuum at angles from the surface normal of 15°, 30°, 45°, 60° and 75°. Spectra were referenced to the reflectivity of an aluminium mirror. Best results were obtained from carefully cut and polished samples while poorer quality spectra were obtained from cut but not polished samples. Several square centimetres of sample area are needed for this technique. A total of five opal-AG, one opal-AN, 8 opal-CT and one opal-C were measured. One of each opal type was examined

using *s*- and *p*-polarised radiation. The wavenumber dependent phase shift spectrum was determined by applying the Kramers-Kronig transform to the 15° spectra using the spectrometer software (OPUS V7.2, Bruker). The complex refractive index, $\eta = n + ik$, and the complex dielectric constant, $\varepsilon = \varepsilon_1 + i\varepsilon_2$, spectra were subsequently calculated from the phase shift and reflectance spectra using the equations given by [13]. The Kramers-Kronig transform was performed between 150 and 2500 cm$^{-1}$ since these boundaries are far from any absorption features and resulted in consistently stable spectra.

**Table 1.** Sample locality data.

| Sample | Opal Type | Locality | Comments |
|---|---|---|---|
| CP-2 | AG | Coober Pedy, South Australia, Australia | |
| G1419 | AN | Mt Cora, New South Wales, Australia | |
| G1442 | AG | William Creek, South Australia, Australia | |
| G7532 | AG | Queensland | |
| G7975 | CT | Mexico | |
| G8608 | AG | White Cliffs, New South Wales, Australia | |
| G8877 | AN | Dalby, Queensland, Australia | |
| G9260 | AG | Australia | |
| G9590 | AG | Coober Pedy, South Australia, Australia | |
| G9594 | AG | Coober Pedy, South Australia, Australia | |
| G9812 | AG | Coober Pedy, South Australia, Australia | |
| G9942 | CT | Angaston, South Australia, Australia | |
| G9887 | CT | Curdimurka, South Australia, Australia | |
| G13740 | CT | Monto, Queensland, Australia | |
| G13743 | CT | Siebengebirge, Germany | |
| G13767 | AG | Lightning Ridge, New South Wales, Australia | Mix of samples |
| G13771 | AG | White Cliffs, New South Wales, Australia | |
| G14581 | AG to CT | Andamooka, South Australia, Australia | Transitional form |
| G15584 | AG | White Cliffs, New South Wales, Australia | |
| G21471 | A | Rotorua, New Zealand | Geyserite |
| G25374 | CT | Somalia (possibly Ethiopia) | |
| G32226 | CT | St Austell, England | |
| G32925 | CT | Zelinograd, Qazaqstan | |
| G32740 | AN | Valec, Czechia | |
| G32752 | CT | Afar, Ethiopia | Mix of samples |
| G33912 | CT | Acari, Peru | |
| G34475 | A | Dubnik, Slovakia | Hydrophane |
| G NEW01 | A | Caldes de Malvella, Spain | Menilite |
| G NEW04 | CT | Mt Lyobo, Tanzania | |
| G NEW18 | CT to C | Oregon, USA | Transitional form |
| G NEW22 | AG | Lambina, South Australia, Australia | |
| G NEW23 | A | Caldes de Malvella, Spain | Menilite |
| M5081 | C | Iceland | |
| MS-1 | AN | Mt Squaretop, Queensland, Australia | |
| MS-4 | AN | Mt Squaretop, Queensland, Australia | |
| OOC4 | AG to CT | Mazaron, Spain | Transitional form |
| T1664 | C | Guanajuanto Mexico | |
| T1665 | A | Rotorua, New Zealand | Geyserite |
| T2222 | CT | Mt Barker, South Australia, Australia | |
| T22824 | AG to CT | Megyasro, Hungary | Transitional form |

Diffuse reflectance spectra (three samples each of opal-AG, opal-AN and opal-CT measured from 700 to 4000 cm$^{-1}$) were obtained as 5 wt% suspensions in KBr using the A562 Integrating Sphere attachment in atmosphere for the Bruker Vertex v80. This device also allows the measurement of a zero-degree angle of incidence diffuse reflectance spectra by placing a flat sample at an alternative port. This is probably the easiest technique to use and only requires a sufficient area of reasonably flat surface.

ATR spectra were measured on the Bruker Vertex 80v under vacuum (far- and mid-range in a consecutive runs) or a Nicolet Nexus 8700 instrument in atmosphere with the Smart Orbit (400–4000 cm$^{-1}$) or Durascope (550–4000 cm$^{-1}$) stages. All devices used a diamond ATR crystal. Since the ATR penetration depth is proportional to wavelength, a simple ATR depth correction was applied to all ATR spectra by multiplying the absorbance values by the wavenumber. No "enhanced" corrections for the refractive index or the angle of incidence were performed.

Powdered ATR samples were prepared without solvent. In the previous work, we used the XRD samples, which were routinely prepared through grinding in acetone, though we believe that this might cause dehydration. Three degrees of grinding were examined. In the first the sample was lightly crushed and introduced onto the diamond plate. Upon lowering of the pressure clamp further crushing occurred as evidenced by movement of material and audible effects. Next, the remaining sample was lightly ground in a tungsten carbide mortar and pestle. Again, some further crushing was apparent when placed on the sample stage. Finally, the residue was extensively ground yielding a fine powder that clumped and seemed unaltered upon placing on the diamond plate. Duplicate runs were made and repeated after the powdered samples had been stored for at least six months. More than six each of opal-AG, opal-AN and opal-CT were analysed with two of opal-C. Polarisation experiments were conducted using the Bruker 80v instrument. Several samples with relatively flat surfaces were also examined by placing the flat surface directly on the ATR plate.

Four 1 g samples were also finely ground dried over a large excess of silica gel for 7 days and then heated in an oven at 80 °C for 7 days. Typical mass losses were 1–2% for each phase. There were no obvious significant differences in mid-IR ATR spectra or XRD patterns. XRD powder diffraction patterns (Bruker D8 Advance machine, Co source K$_\alpha$ = 1.78897 Å) were recorded with a scan speed of 0.0195° per second over the 2$\theta$ range 10 to 65° [7].

Scanning electron microscopy (SEM) images were obtained using a FEI Inspect 50 SEM (FEI, Hillsboro, OR, USA) by recording the secondary electron emission at an acceleration voltages of either 5 or 10 kV and working distances between 4 and 6 mm (actual values used were based on image quality). The sample was sputter coated with a 2 nm layer of platinum to reduce charging. No etching was necessary.

Raman spectra in the region 600–1500 cm$^{-1}$ were recorded with a XploRA Horiba Scientific confocal Raman system (Horiba, Kyoto, Japan) using a ×50 objective (numerical aperture 0.6), 786 nm laser (27 mW) and a spectrometer resolution of ~4.5 cm$^{-1}$ full width at half height (FWHM). Opal is a very weak Raman scatterer in the region of interest and thus required 4 h of data collection for each sample.

Finally, we note that most samples were selected on the basis of showing little or no quartz in their XRD patterns. Our experience, however, is that even traces of quartz give strong XRD patterns but when checked by Raman spectroscopy show little quartz is actually present. We have, however, explored some specimens that show additional reflections in the XRD pattern due to impurities (see Appendix B).

## 3. Overview of Previous Silica IR Studies

While previous examination of opal has been limited, there is extensive literature on silica including glasses. The three bands described above are common features for opal and, with comparison to glasses, they may be split into transverse optical (TO) and longitudinal optical (LO) components (see [14,31,32] for discussion of the phenomenon). TO phonons create an oscillating dipole moment that is perpendicular to the phonon propagation direction, whereas the dipole moment for LO phonons oscillate in the propagation direction. Long-range electrostatic interactions cause the energies of the LO and TO phonons to diverge creating LO-TO splitting. For normal incidence on a surface, only TO vibrations can be excited with infrared light, whereas oblique incidence allows excitation of both TO and LO vibrations. This is referred to as the Berreman effect for thin films [33] and has

been observed for bulk silica glasses [14]. Both TO and LO modes are potentially visible in the Raman spectrum for non-centrosymmetric samples, and LO bands occur at higher energies than the TO bands as required by the Lyddane–Sachs–Teller relation.

We use the nomenclature (e.g., [15]) of $TO_1/LO_1$, $TO_2/LO_2$ and $TO_3/LO_3$ to assign the bands in the 400–470, 790–800 and 1000–1300 $cm^{-1}$ regions, respectively. The $LO_3$ band is readily apparent in the IR spectrum for vitreous glass (e.g., [15,17,34]) while all of $LO_1$, $LO_2$ and $LO_3$ bands are present in Raman spectra [12]. The $LO_1$ and $LO_3$ bands are at higher energy than the corresponding TO bands while $TO_2$ and $LO_2$ bands are very close together [13]. In addition to these bands, other weaker transitions have been observed in the infrared spectrum. We note that assignment of opal-A and opal-CT peaks is incomplete, though specific infrared and Raman frequencies related to defined silica species such as quartz, cristobalite and tridymite are well documented [30,35–37]. For instance, the characteristic opal-A peak or shoulder at approximately 550 $cm^{-1}$ is unassigned, though this may be due to bond bending [11,38]. Bands due to silanol groups (non-bridging oxygen) will be expected in silica species with Si-OH stretches at 860 $cm^{-1}$ [16], 940–960 $cm^{-1}$ [39], 950–960 $cm^{-1}$ [21], 967 $cm^{-1}$ [40] and 985 $cm^{-1}$ [16] reported. This may be affected by H-bonding. Between the $TO_3$ and $LO_3$ positions, there is a region of high reflectivity which is termed the *Reststrahlen* peak [16], while a reduction in reflectance due to the Christiansen effect (when the refractive index approaches unity) [16] is seen for opal-A. $TO_4/LO_4$ bands have been proposed [15], though their nature is uncertain. Modelling of conformations of silica led to identification of discrete $TO_3/LO_3$ bands for 4- and 6-membered silicate rings [41,42]. It has been proposed that water librational bands coincide with the medium-energy $TO_2/LO_2$ absorptions [25].

Table 2 gives a compilation of literature sources though it should be noted that actual position of transmission, ATR and reflectance measurements of the same compound will produce different spectra, including displaced peak positions. For instance, ATR leads to lower-frequency peak positions when compared with the transmission mode and the peaks themselves are distorted compared to transmission. Kramers-Kronig (specular reflectance) or Kubelka-Munk (diffuse reflectance) correction algorithms may be used to compare the different spectral outputs. While this may be carried out using in-built instrument software, such processes may be opaque [43], hence comparison should be treated with caution, especially for results from different studies, particularly if corrections are made. We also note differences in terminology in previous works when describing vibrational modes.

For reference, Raman spectra of opal and silica [7,30,36,37,44,45] show that the major peaks for opal-A and opal-CT in the 200–500 $cm^{-1}$ region tend to be broad with varying degrees of fine detail (see Figure 1 in [7]). It has been proposed that Raman peaks at 495, 820 and 1200 $cm^{-1}$ in vitreous silica are due to LO modes [12,46]. Opal-C, on the other hand, shows relatively sharp peaks similar to those for $\alpha$-cristobalite (e.g., Figure 1 in [37]).

**Table 2.** Literature review of assigned IR features seen for silica species, grouped by band (see text).

| Frequency ($cm^{-1}$) | Description | Substrate | References |
|---|---|---|---|
| 457–490 | $TO_1$: Si-O-Si rock | Silica, opal-A and glass | [13,15–17] |
| 507 | $LO_1$ Si-O-Si rock | Silica and glass | [13,15] |
| 784–810 | $TO_2$; Si-O-Si symmetric stretch | Silica and glass | [13,15,17] |
| 820 | $LO_2$; Si-O-Si symmetric stretch | Silica and glass | [13,15] |
| 870 | H-bonded Si-OH stretch | Opal-A | [16] |
| 870 | Non-bridging oxygen vibration | Silica | [47] |
| 950–985 | Isolated SiOH stretch | Opal-A | [9,16,40–42] |
| 1040–1260 | $TO_3$: Si-O-Si antisymmetric stretch | Xerogel, glass, opal-A, silica | [13,15–17,34,41,42] |
| 1111 | Antisymmetric stretch (*Reststrahlen*) | Opal-A | [16] |
| 1130–1286 | $LO_3$: Si-O-Si antisymmetric stretch | Xerogel, glass | [13,15–17,34,41,42] |
| 1170 | $LO_4$: disorder-induced mode | Silica | [15] |
| 1200 | $TO_4$: disorder-induced mode | Silica | [15] |
| 1266 | Antisymmetric stretch (*Reststrahlen*) | Opal-A | [16] |
| 1361 | Christiansen feature | Opal-A | [16] |

## 4. Results

### 4.1. SEM Examination

Extensive SEM examinations were carried out to eliminate the possibility that heterogeneity caused variability in ATR measurements. For instance, layers of different types of silica, e.g., cristobalite and tridymite [3], or areas having differing ratios of $Q_4$ (Si atoms bonded to four bridging oxygen atoms) to $Q_3$ (Si atoms bonded to three bridging oxygen atoms and one hydroxyl) might be exposed on grinding of opal-AG, though as Figure 2 shows this is unlikely to be the case with only minor disruption observed. The initial phase of crushing gave pieces in the 200–600 µm size range while light grinding led to material from 10 µm diameter up to those seen for the light crushing. The most ground samples had a different visual appearance with considerable agglomeration of particles, typically in the range 4–80 µm. The most ground samples also showed signs of some detritus amorphous silica (e.g., Figure 2d). EDX analysis showed this to be mostly Si and O with a small amount of Al. This contrasts with the spherical regions that also showed the presence of Na and K (greater than 1 wt%). At most, this detritus material covered 10% of the scanned area. The origin of this material is unknown, though a reasonable proposal is that it represents the intersphere material that has become desiccated in the SEM process. No obvious analogues were seen in the ground samples of opal-AN, opal-CT and opal-C. The SEM images of opal-CT (Figure 3) again show a variety of types (also see [4,48]) though with no evidence for any changes upon grinding.

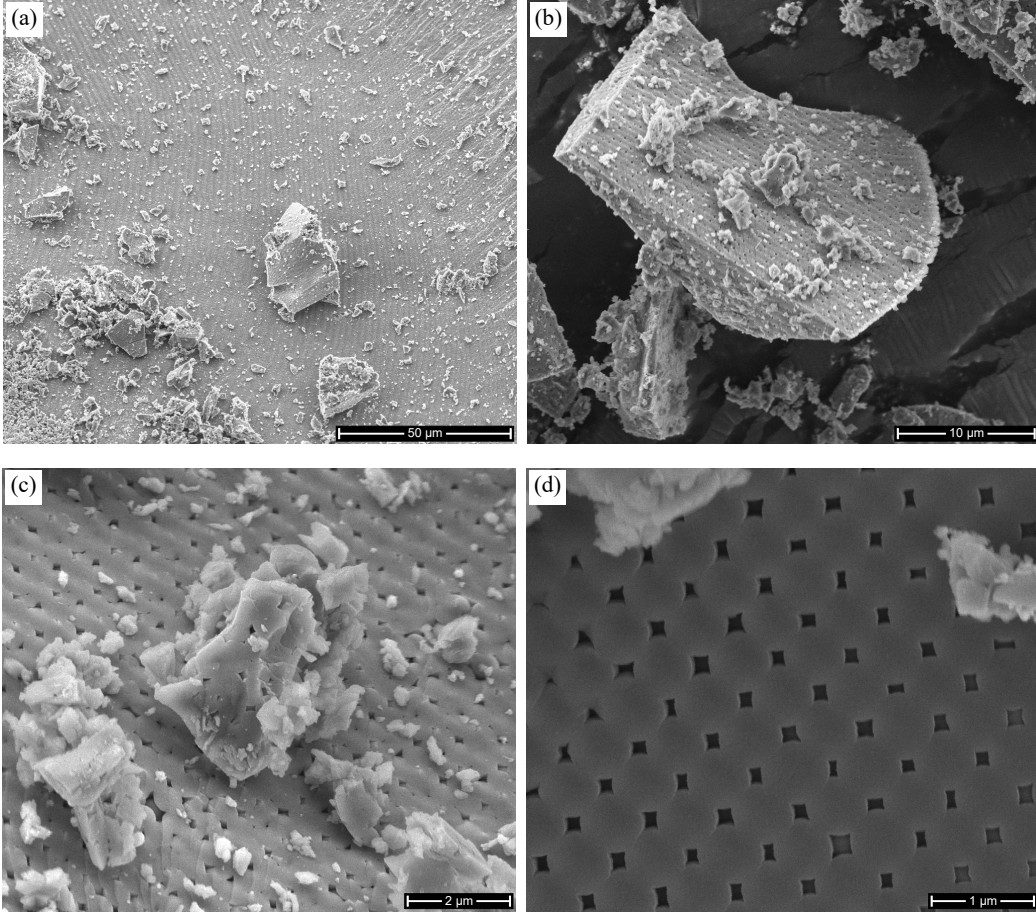

**Figure 2.** SEM images showing the effect of grinding on a sample of opal-AG (G9812 from Coober Pedy South Australia, Australia) (**a**) lightly crushed and (**b**–**d**) finest grinding with increasing magnification.

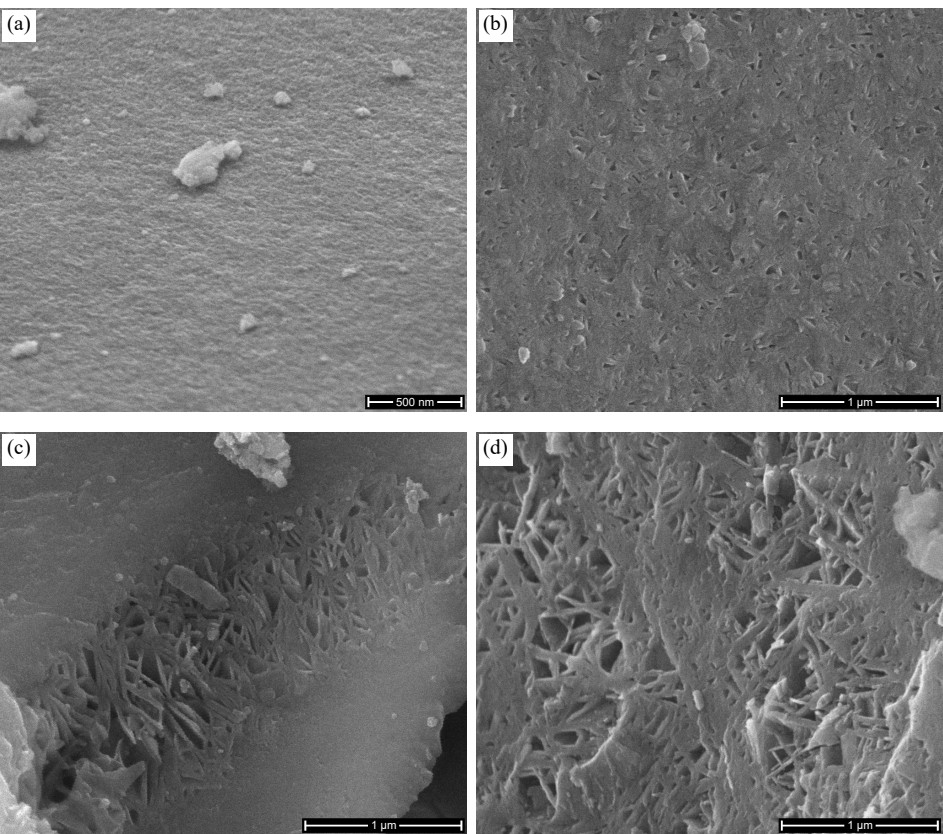

**Figure 3.** SEM images showing differing appearances of selected zones of opal-CT: (**a**) G25374 (labelled as from Somalia), (**b**) G7975 fire opal from Mexico, (**c**) G13740 opalized wood from Monto, Queensland, Australia, and (**d**) G9942 from Angaston, South Australia, Australia.

*4.2. Transmission Spectra*

Full-range transmission spectra may be obtained using CsI as the supporting medium, whereas KBr and KCl may suffer extinction problems at lower energies. Consistent spectra were seen for each opal type, with typical examples shown in Figure 4. The three bands (Table 3) correspond to those previously found in powder ATR [7] and other transmission mode studies [9,20–22,25,26]. The differentiating features of peaks at approximately 550 cm$^{-1}$ for opal-A and 620 cm$^{-1}$ for opal-C are again seen. The shoulders at approximately 1220 cm$^{-1}$ are presumably due to the LO$_3$ bands while those at approximately 510 cm$^{-1}$, observed for some of the opal-CT samples (see below), may indicate a contribution from LO$_1$. The shoulders at approximately 950 cm$^{-1}$ are consistent with the presence of silanol.

Peak positions show credible differences between the opal types (Table 3). The shifted position and broadness of the medium-energy band for opal-A when compared to opal-CT and opal-C is very apparent in this mode. Similar results were obtained using KBr as the supporting medium though with change in position possibly due to refractive index effects. For instance, the medium-energy peaks are shifted to lower energy in KBr but retain their relative position (opal-A 793 ± 1 cm$^{-1}$ and opal-CT 787 ± 2 cm$^{-1}$). Spectra recorded in both CsI and KBr matrices display the large difference between baseline levels on either side of the higher-energy band. Where tried, KCl discs gave poor spectra.

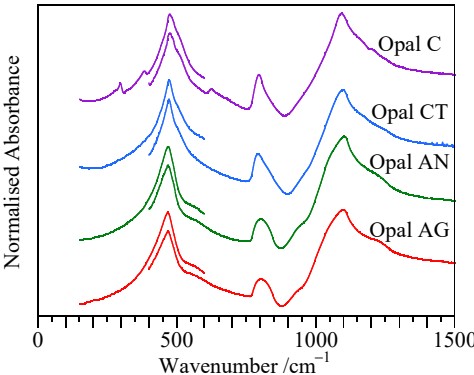

**Figure 4.** Transmission spectra of examplars prepared as a suspension in CsI disc: (red) opal-AG (G13771 from White Cliffs, Australia), (green) opal-AN (G32740 hyalite from Valec, Chechia), (blue) opal-CT (G9942) from Angaston, Australia) and (purple) opal-C (M5081 from Iceland). Spectra have been normalised to the 470 cm$^{-1}$ band. Far- and mid-range IR spectra are from different instruments and are offset for clarity.

**Table 3.** Transmission mode spectra (CsI disc) peak positions. The uncertainty value represents the range of values observed for various samples.

| Sample | Low-Energy Band (Synchrotron) /cm$^{-1}$ | Low-Energy Band (Nicolet) /cm$^{-1}$ | Medium-Energy Band /cm$^{-1}$ | High-Energy Band /cm$^{-1}$ |
|---|---|---|---|---|
| Opal-AG | 469 ± 1 | 470 ± 2 | 802 ± 1 | 1099 ± 2 |
| Opal-AN | 469 ± 1 | 469 ± 1 | 801 ± 1 | 1103 ± 2 |
| Opal-CT | 472 ± 1 | 472 ± 1 | 793 ± 2 | 1099 ± 2 |
| Opal-C | 475 ± 1 | 474 ± 2 | 795 ± 1 | 1095 ± 2 |

### 4.3. Reflectance Spectra

Three different modes were available: variable incidence angle (15–75°) specular reflectance, diffuse reflectance of 5% suspensions in KBr and a zero-degree incident angle diffuse reflectance using the integrating sphere attachment (see Experimental section).

The best overall quality variable incidence angle specular reflectance spectra were gained from the polished specimens (Figure 5) and again the spectra for opal-AG and opal-AN were indistinguishable. The characteristic opal-A peak at 550 cm$^{-1}$ is barely visible while the medium energy peaks are only weak. By analogy with silica and glasses, longitudinal optical (LO$_3$) bands [13–15,17,34,46] appear for all opal types above 1200 cm$^{-1}$. These are more apparent at the lower incidence angles for opal-A than for opal-CT. The LO$_3$ bands increase in intensity with incidence angle as previously noted [14,34,49,50]. At a 75° incident angle this peak is at approximately 1280 cm$^{-1}$ for all types. A further peak appears in the higher-energy region for some opal-CT samples at approximately 1180 cm$^{-1}$. Fused silica shows a similar spectrum to opal-CT also with a delineated peak at 1180 cm$^{-1}$. This peak is variable in intensity and may appear as a separate peak (as for G9942 in Figure 5), or as a flat region between the two higher-energy peaks (see later). A major difference between opal types lies at higher incidence angles with an extra band seen for opal-CT at approximately 510 cm$^{-1}$ which overlaps the TO$_1$ peak at approximately 470 cm$^{-1}$. This is presumed to be due to a LO$_1$ band. The single sample of opal-C examined (Figure 5d) showed complicated effects but LO$_1$ and LO$_3$ bands were likely present. The peak at 620 cm$^{-1}$ is retained. While small, the TO$_2$ peaks do not show significant variation and thus provided little support for the presence of a LO$_2$ band.

Peak positions and shapes of the absorptivity index (Figure 6a, Table 4), derived from applying the Kramers-Kronig transform to the 15° reflectance spectra, are consistent with transmission mode spectra. However, this should be treated as an indicative exercise since

we find that altering the start and end point of the in-built Kramers–Kronig transform algorithm has a slight effect on peak positions and thus the quoted derived values need to be treated with caution. This spread means it is not possible to differentiate opal-A from opal-CT from the low-energy peak as the ranges overlap. The distinctive shapes of the medium-energy bands are, however, retained. Variability within the opal-CT reflectance spectra is discussed later.

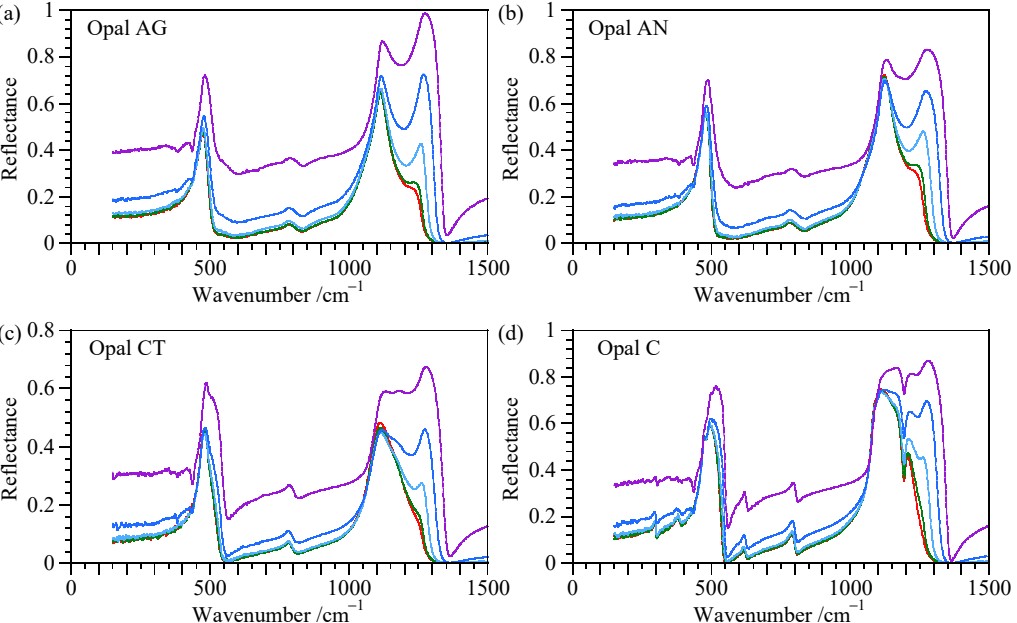

**Figure 5.** Specular reflectance spectra (unpolarised light) of polished thin segments of (**a**) opal-AG (CP-2 from Coober Pedy, South Australia, Australia), (**b**) opal-AN (MS-1 from Mount Squaretop, Queensland, Australia), (**c**) opal-CT (G9942 from Angaston, South Australia, Australia) and (**d**) opal-C (T1664 from Guanajuanto, Mexico) (red 15°, green 30°, light blue 45°, blue, 60° and violet 75°).

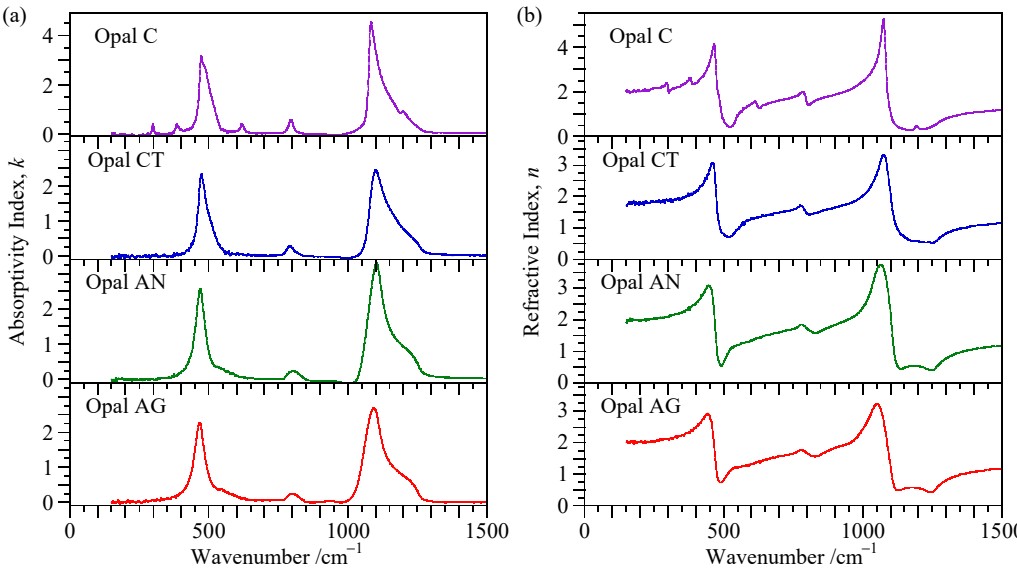

**Figure 6.** Kramers-Kronig-transformed spectra of polished slabs of (red) opal-AG (CP-2 from Coober Pedy, South Australia, Australia), (green) opal-AN (MS-1 from Mount Squaretop, Queensland, Australia), (blue) opal-CT (G9942 from Angaston, South Australia, Australia) and (violet) opal-C (T1664 from Guanajuanto, Mexico)); (**a**) the absorptivity index, $k$, and (**b**) the refractive index, $n$.

**Table 4.** Absorptivity index peak positions derived from the Kramers-Kronig transform of the 15°
incidence angle reflection measurements for the samples shown in Figure 6.

| Sample | Low-Energy Band /cm$^{-1}$ | Medium-Energy Band /cm$^{-1}$ | High-Energy Band /cm$^{-1}$ |
|---|---|---|---|
| Opal-AG | 467 | 802 | 1094 |
| Opal-AN | 469 | 804 | 1103 |
| Opal-CT | 474 | 792 | 1101 |
| Opal-C | 472 | 795 | 1084 |

The refractive index spectrum (Figure 6b) for opal shows large variation according to wavelength, with values ranging from below one to significantly above that for diamond ($n$ = 2.42) which is constant in the IR range [51]. This contrasts with those values for the optical range which are generally in the vicinity of $n$ = 1.4 [39,52] though vary with water content [52,53]. There is large variation between opals within each type which could be due to real differences, lack of surface smoothness or an artefact introduced by the Kramers-Kronig algorithm. Opal may also show birefringence [52] which would affect the reflection spectrum. The figures are, however, in accord with values for silica glass [54] with large figures for the lower- and higher-energy peaks but lower for the medium-energy one. Peaks in the absorptivity and refractive index spectra are coincident.

The real and imaginary components of the complex dielectric constant allow delineation of the TO and LO bands [13]. Peaks in Im{$\varepsilon$} correspond to TO bands whilst peaks in the energy loss function, Im{$-1/\varepsilon$}, correlate with LO bands. Figure 7 shows the Im{$\varepsilon$} and Im{$-1/\varepsilon$} spectra, and peak maxima are given in Table 5. This clearly demonstrates the presence of TO and LO components of all bands. Originally, it was suggested that the major peak at 1250 cm$^{-1}$ in the Im{$-1/\varepsilon$} spectra may be an artefact with the real LO$_3$ band at 1200 cm$^{-1}$ [13] but more recent work (e.g., [15]) gives support for the peak being the LO$_3$ band. The LO$_1$ band relative to the TO$_1$ band appears less intense for opal-A than for opal-CT and opal-C. Again, it is noted that these are from manipulated data, so actual peak position is uncertain. The relative intensities of the TO$_1$ and LO$_1$ and their separation may explain the greater prominence of the 510 cm$^{-1}$ shoulder in reflectivity spectra for opal-CT in contrast to opal-A. TO$_2$ and LO$_2$ bands are nearly coincident. It is not clear whether the shoulders at approximately 1200 cm$^{-1}$ for opal-A and opal-CT represent incomplete resolution or the presence of additional bands, or composite silica species. It has been suggested that in thin silica films, TO$_4$ and LO$_4$ bands at 1200 and 1170 cm$^{-1}$, respectively, are present in this region and have been ascribed to disorder-induced modes [15], although this would mean a reversal of the usual TO-LO ordering.

Specular reflection spectra are expected to show differences when polarised radiation is used for the measurement. This is demonstrated by using the Fresnel equations and the complex refractive index determined from Kramers-Kronig-transformed data to calculate the expected reflectance spectra for *s*- and *p*-polarised light at a 75° angle of incidence. Figure 8a shows the predicted spectra for the samples examined in Figure 6. The calculations clearly demonstrate that there is a significant difference in the ratio between the LO$_3$ and TO$_3$ bands when using *s*- and *p*-polarisations. Thus, calculations predict that with *p*-polarised light, LO$_3$ bands become more dominant than the TO$_3$. Likewise, the LO$_1$ (where apparent) is predicted to be enhanced by *p*-polarised light. In the LO$_2$/TO$_2$ band region, the peak is calculated to be diminished with *p*-polarisation. However, it should be noted that the band intensity is affected by baseline effects. Measured spectra using polarised incident radiation at a 75° angle of incidence are shown in Figure 8b and display an overall shape consistent with that predicted, albeit with reduced absolute reflectance. Predictions and measurements using polarised light at a 15° angle of incidence were also performed and showed that the *s*- and *p*-polarised spectra were expected to be very similar except for slight intensity differences for the LO$_3$ band. A similar effect whereby the LO$_3$ band becomes enhanced with *p*-polarisation has been reported for soda lime glass [55]. We

have also observed this polarisation dependent spectra in flat sample ATR measurements and we discuss this further in Section 4.5.

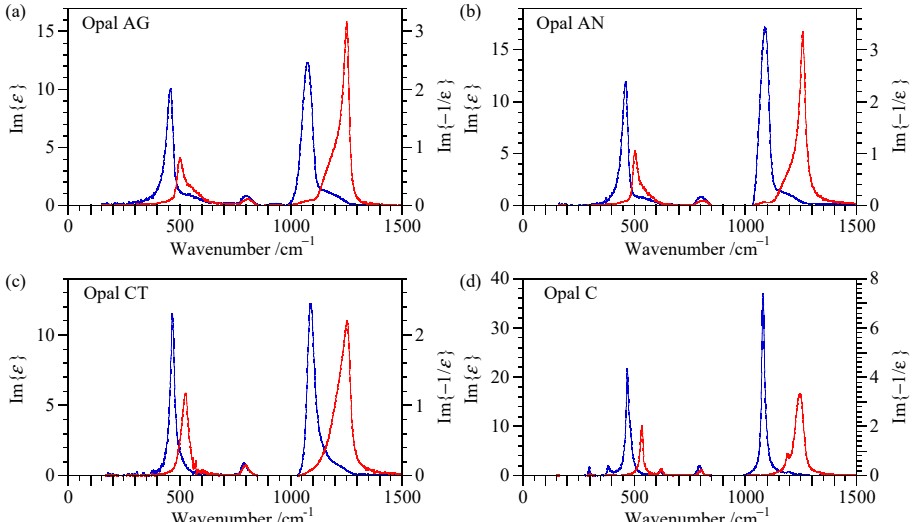

**Figure 7.** Im$\{\varepsilon\}$ (blue) and Im$\{-1/\varepsilon\}$ (red) spectra derived from a Kramers-Kronig transform of polished slabs showing TO and LO components, respectively. Samples are (**a**) opal-AG (CP-2 from Coober Pedy, South Australia, Australia), (**b**) opal-AN (MS-1 from mount Squaretop, Queensland, Australia), (**c**) opal-CT (G9942 from Angaston, South Australia, Australia) and (**d**) opal-C (T1664 from Guanajuanto, Mexico). Note that the Im$\{-1/\varepsilon\}$ spectrum is displayed 5× enhanced compared to the Im$\{\varepsilon\}$ spectrum. Im$\{\varepsilon\}$ values are given on the left axis and Im$\{-1/\varepsilon\}$ values are displayed on the right axis.

**Table 5.** TO and LO band peak positions taken as the maxima in the Im$\{\varepsilon\}$ and Im$\{-1/\varepsilon\}$ spectra that were derived from the Kramers-Kronig transform of 15° incidence angle reflection spectra for samples displayed in Figure 6.

| Sample | Low-Energy Band TO$_1$/LO$_1$ /cm$^{-1}$ | Medium-Energy Band TO$_2$/LO$_2$ /cm$^{-1}$ | High-Energy Band TO$_3$/LO$_3$ /cm$^{-1}$ |
|---|---|---|---|
| Opal-AG | 458/503 | 799/807 | 1080/1253 |
| Opal-AN | 463/505 | 800/810 | 1091/1258 |
| Opal-CT | 468/529 | 790/798 | 1090/1254 |
| Opal-C | 470/536 | 793/802 | 1080/1245 |

The zero-degree angle of incidence reflectance spectra measured with an integrating sphere (Figure 9a) were similar to those obtained at 15° using the specular reflectance apparatus. Importantly, these were conducted in air while the specular reflectance spectra were measured under vacuum. Consistency suggested no major dehydration effects in the region of interest. The most obvious feature was that the LO$_3$ band for opal-A (at approximately 1240 cm$^{-1}$) was more prominent than for opal-CT.

Kubelka–Munk-corrected diffuse reflectance spectra (Figure 9b) were similar to those for transmission and specular reflectance with peaks at 797, 800 and 789 cm$^{-1}$ for the medium-energy bands for opal-AG, opal-AN and opal-CT, respectively. The peak for opal-A was again at higher energy and broader than for opal-CT. Clear differences were seen in the mid-section (presumed to be silanol), with shallow peaks visible at approximately 945, 955 and 970 cm$^{-1}$ for opal-AG, opal-AN and opal-CT, respectively. Curve fitting using Gaussian line shapes resolved these to 937, 948 and 957 cm$^{-1}$, respectively, with FWHM of approximately 55 cm$^{-1}$ in all cases. Similar fitting of the TO$_3$/LO$_3$ region showed no significant differences between the opal types, with average peak positions of 1089 ± 8 and 1203 ± 6 cm$^{-1}$. Equivalent FWHM were 121 ± 9 and 95 ± 6 cm$^{-1}$. There were no differences with polarised radiation.

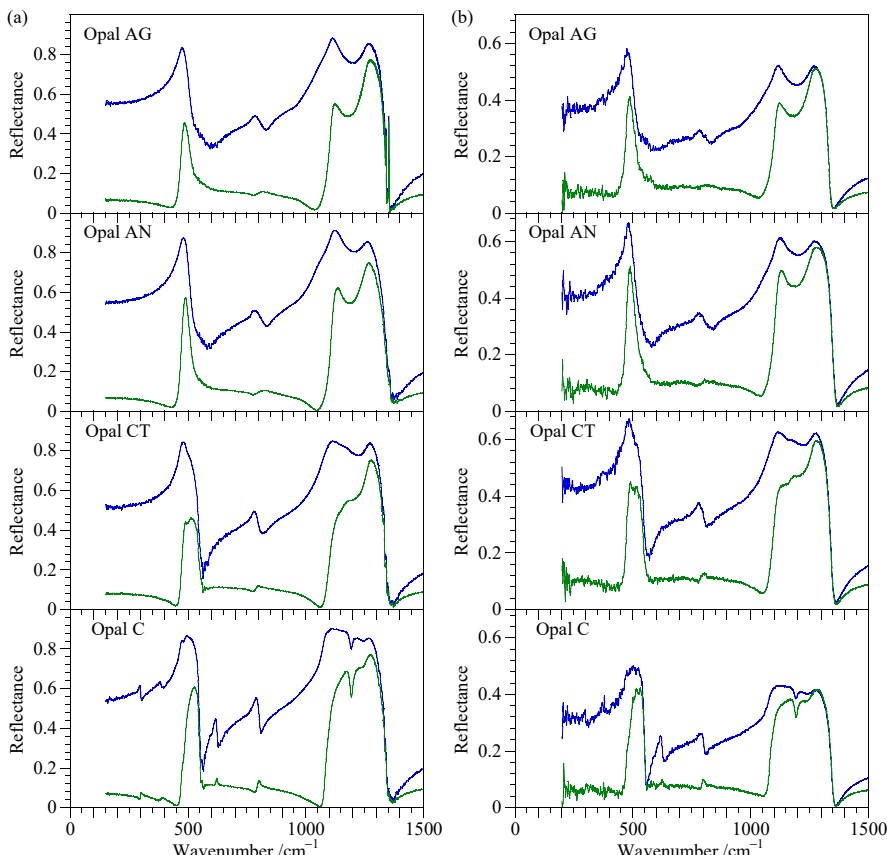

**Figure 8.** Polarised 75° angle of incidence specular reflectance spectra of polished thin segments. (**a**) Predicted spectra calculated using the complex refractive index determined from 15° measurements and (**b**) measured spectra. From top to bottom, the samples are opal-AG (CP-2 from Coober Pedy, South Australia, Australia), opal-AN (MS-1 from Mount Squaretop, Queensland, Australia), opal-CT (G9942 from Angaston, South Australia, Australia) and opal-C (T1664 from Guanajuanto, Mexico). Spectra in green represent *p*-polarised spectra and those in blue are *s*-polarised.

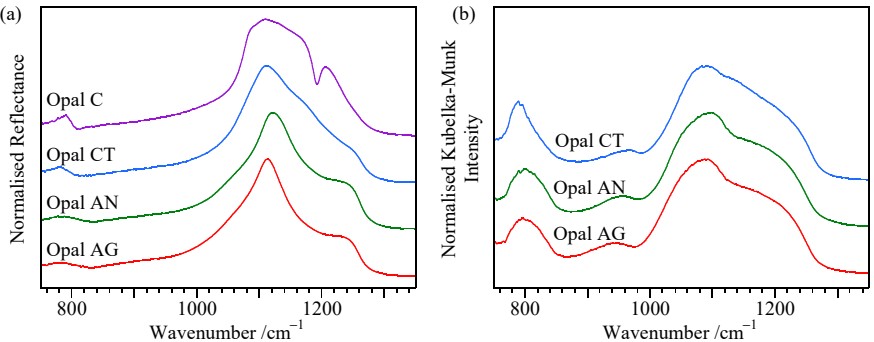

**Figure 9.** Reflectance spectra recorded with an integrating sphere attachment for the Bruker v80 instrument. (**a**) Zero-degree angle of incidence spectra of flat opal samples (red): opal-AG (CP-2 from Coober Pedy, South Australia, Australia), (green): opal-AN (MS-1 from Mount Squaretop, Queensland, Australia), (blue): opal-CT (G9942 from Angaston, South Australia, Australia) and violet): opal-C (T1664 from Guanajuanto, Mexico). (**b**) Kubelka-Munk-corrected diffuse reflectance spectra of 5 wt% suspensions in KBr (red): opal-AG (G9590 from Cooper Pedy, South Australia, Australia), (green): opal-AN (MS-4 from Mount Squaretop, Queensland, Australia) and (blue): opal-CT (G9887 honey opal from near Curdimurka, South Australia, Australia). Spectra have been offset to aid comparisons.

### 4.4. ATR Measurements of Ground Samples

ATR spectra of each sample were measured at least six times with two spectra taken for each grinding state (see Experimental section). This was necessary because significant measurement-to-measurement variation was seen even for repacking of the same material on the diamond plate. Two general features were noted. First, the variation mostly affected the peaks at approximately 470 and 1100 cm$^{-1}$ with respective peak position spreads of approximately 20 and 40 cm$^{-1}$. Those for the 790 cm$^{-1}$ band were less than 10 cm$^{-1}$. Second, there was a general trend whereby the peaks at 470 and 1100 cm$^{-1}$ were shifted to lower energy and became broader upon more grinding. This is shown in Figure 10 though it should be noted that these data have been selected to show the effect as there are major measurement-to-measurement differences. The differences in the position and shape of the 790 cm$^{-1}$ peak for opal-A and opal-CT, seen in transmission and specular reflectance modes, are repeated here. The same observations were seen with samples run on both the Nicolet and Bruker FTIR instruments. Fused silica and Gilson synthetic opal showed similar effects when ground.

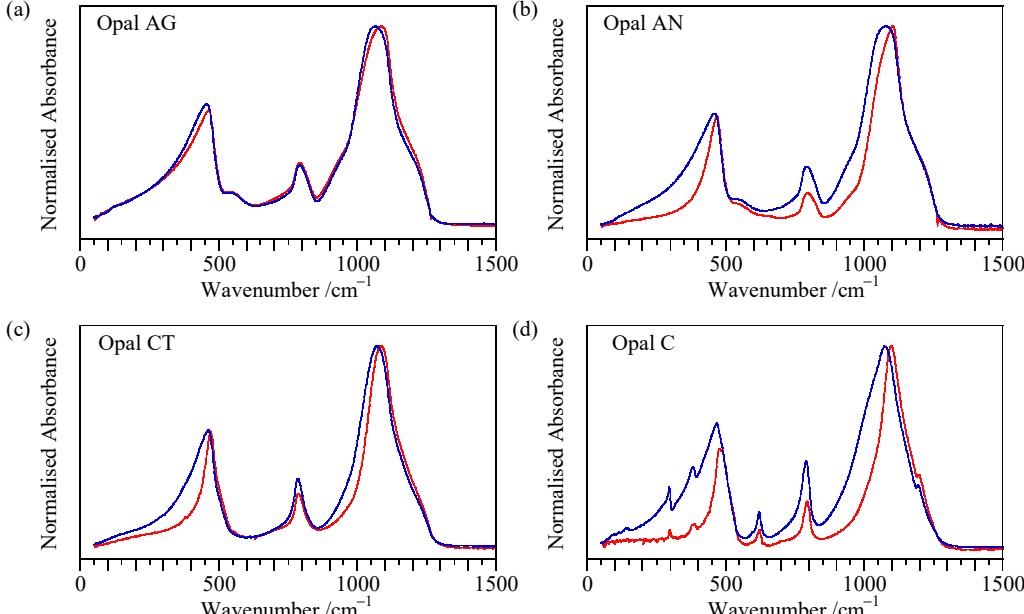

**Figure 10.** ATR spectra (Bruker 80v instrument) of ground samples showing selected examples of variable spectra with grinding (red less ground with blue more ground): (**a**) opal-AG (G NEW22 from Lambina, South Australia, Australia), (**b**) opal-AN (hyalite) (G8877 from Squaretop, Dalby, Queensland, Australia), (**c**) opal-CT (G9942 from Angaston, South Australia, Australia) and (**d**) opal-C (T1664 from Guanajuanto, Mexico). Spectra are depth corrected and normalised to the 1100 cm$^{-1}$ peak.

Correlation between the peak maxima for the 470 and 1100 cm$^{-1}$ peaks is shown in Figure 11 and demonstrates a consistent effect. Samples from the lower left of the diagrams (i.e., samples with peaks at lower energy) all show a significant tail in the far-IR as seen in the finely ground spectra of Figure 10. The data in Figure 11 also show that there is a slight difference in the peak maxima between opal-A and opal-CT as was also observed in transmission mode spectra. While the correlation for the 790 cm$^{-1}$ is less convincing, possibly owing to the low overall range, the differences in peak position are stark. The characteristic peak at 620 cm$^{-1}$ for opal-C appeared unchanged with grinding.

There were no obvious differences found with extended standing at room temperature, dehydration or heating at 80 °C. No obvious differences were seen between the XRD patterns of the samples ground to a different degree or when compared with samples ground in acetone.

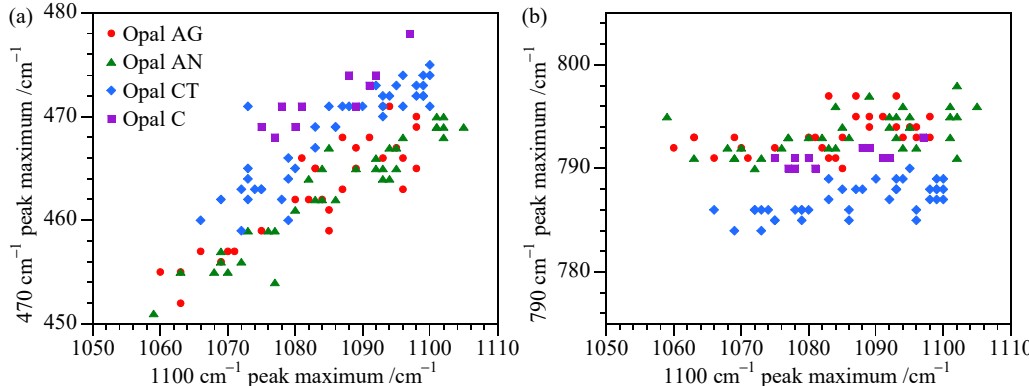

**Figure 11.** Correlation of peak positions for the powder ATR spectra: (**a**) 470 and 1100 cm$^{-1}$ peaks and (**b**) the 790 and 1100 cm$^{-1}$ peaks. Peak positions are taken from spectra that have been depth corrected.

We have attempted to deconvolute the medium and higher wavenumber band into constituent components and although individual spectra could be fit well, there was no consistency in peak areas, widths, and positions between the different measurements. In particular, we tried to determine whether the mid-range peak for opal-A is composite (e.g., TO$_2$ and LO$_2$) as its shape indicates. While it could be deconvoluted into two peaks for any particular spectrum it was not possible to determine a consistent pattern of a combination of two fixed peaks, so the conjecture remains unanswered. Polarisation effects are observed in the ATR spectra, as shown in Figure 12. In particular, for TO$_1$/LO$_1$ and TO$_3$/LO$_3$ bands, *p*-polarised light gives greater absorption and shifts the peak maximum to higher energies. Conversely, the TO$_2$/LO$_2$ band remains unshifted. These types of effects have been reported previously for quartz and silica films [47].

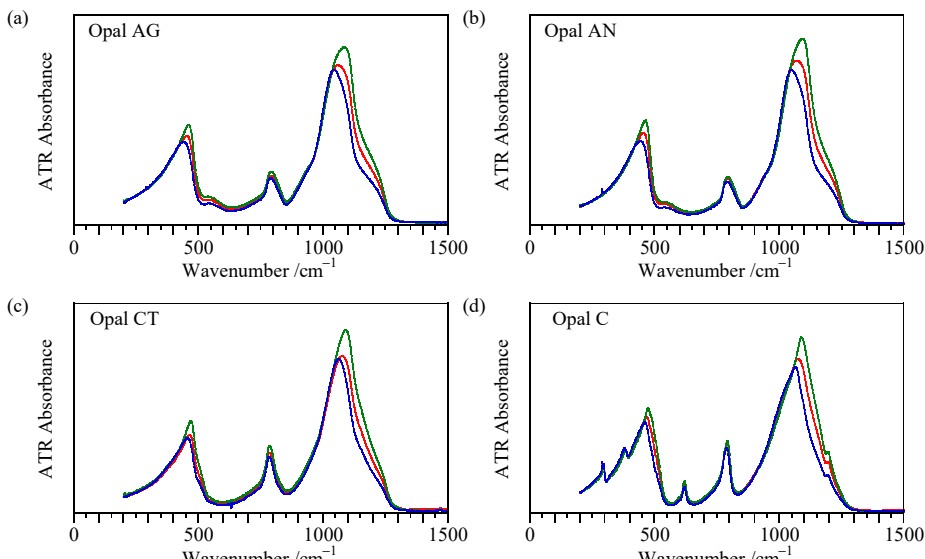

**Figure 12.** ATR spectra of the most finely ground samples showing the effect of polarisation on (**a**) opal-AG (G NEW22 from Lambina, South Australia, Australia), (**b**) opal-AN (G8877 from Squaretop, Dalby, Queensland, Australia), (**c**) opal-CT (G9942 from Angaston, South Australia, Australia) and (**d**) opal-C (T1664 from Guanajuanto, Mexico). Spectra are depth corrected but not otherwise manipulated. Red is unpolarised, green is *p*-polarised and blue is *s*-polarised.

### 4.5. ATR Measurement of Flat Surfaces

ATR measurement of flat surfaces represents an additional IR technique for a standard ATR attachment. Put simply, the sample is merely placed on the ATR diamond plate and

if there is sufficient contact a spectrum is obtained. Typically, such a technique would only be used for a soft substrate, such as rubber, where powder ATR is not a viable option. Additionally, special preparation of the samples may be needed to achieve a sufficiently flat surface for good contact to be made with the diamond plate. Thus, this is a specialised rather than standard usage. As far as we are aware, there is no literature precedence for comparison of the two ATR methods for silica. As will be seen, although spectra are distinct and reproducible, the theoretical understanding is challenging.

The best quality spectra were obtained from polished flat samples. While superficially similar to the other measurement methods, the spectra (shown in Figure 13) are distinct. For spectra recorded with unpolarised light, four features are noteworthy. First is that both the $TO_1/LO_1$ and $TO_3/LO_3$ bands have large shifts to lower energy—typically to 400 and 1000 cm$^{-1}$ respectively. Opal-CT samples gave slightly higher peak positions (420 cm$^{-1}$) compared with 400 cm$^{-1}$ for opal-AG. This delineates the characteristic peak approximately 550 cm$^{-1}$ for the opal-AG and opal-AN samples. Second is that the peak at 790 cm$^{-1}$ is relatively fixed in position but that it has increased in relative intensity when compared to powder ATR spectra. Third is that spectra for opal-AG, opal-AN, opal-CT and opal-C are again different. Fourth, additional peaks appear above 1100 cm$^{-1}$ presumably due to $LO_3$ contributions. These peaks are extremely variable in intensity and probably differ according to sample presentation (e.g., flatness and orientation on the sample) and the instrument (e.g., plate size and flatness, and angle of incidence of incoming radiation). Merely moving a sample a few millimetres often caused the higher-energy region to change markedly with the absence or presence of a $LO_3$ band. Striking examples of positional variability are shown in Appendix A (Figure A1) when spectra were recorded using an ATR unit with a cracked diamond stage on the Nicolet instrument with a Smart Orbit attachment. This will have induced differences in the degree of contact between the sample and diamond plate. Finally, all samples are affected by polarisation again suggesting reflection-like behaviour at least for the low and higher-energy regions. Despite the larger medium-energy peak intensity, it was little affected by polarisation. The isosbestic points at approximately 1030 cm$^{-1}$ are in a similar position to those for powder ATR.

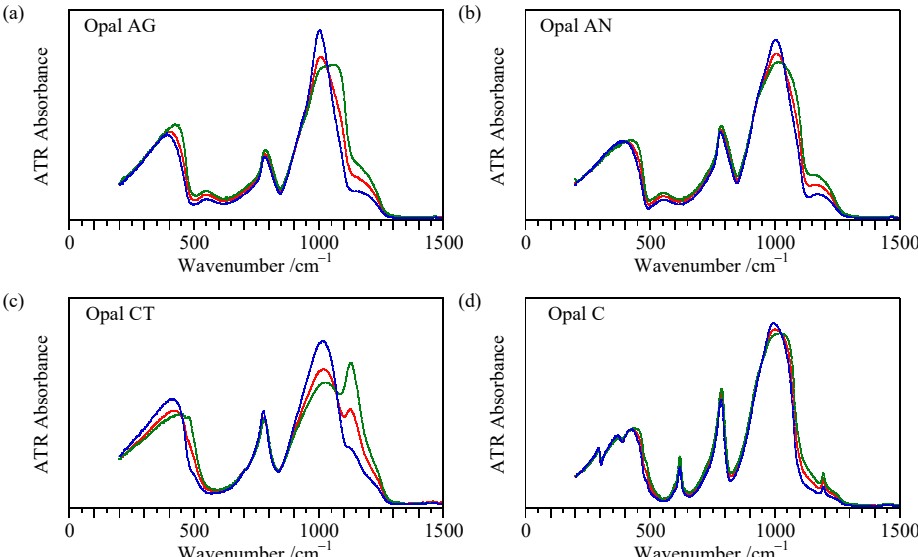

**Figure 13.** Flat sample ATR measurements (Bruker v80 instrument) of polished thin slice showing the effect of polarisation (**a**) opal-AG (CP-2 from Coober Pedy, South Australia, Australia, (**b**) opal-AN (MS-1 from Mount Squaretop, Queensland, Australia, (**c**) opal-CT (G9942 from Angaston, South Australia, Australia) and (**d**) opal-C (T1664 from Guanajuanto, Mexico). Red is unpolarised, green is *p*-polarised and blue is *s*-polarised. IR spectra are corrected for ATR depth penetration but not otherwise manipulated.

### 4.6. Calculation of Si-O-Si Bond Angles

It has been reported that Si-O-Si bond angle ($\theta$) may be calculated using the low-medium-energy peak position to determine the $\beta$ and $\alpha$ terms, respectively, using central force theory [17,56,57], as follows:

$$low\ energy\ peak\ position = \frac{1}{2\pi c}\sqrt{\frac{2\beta}{m_O}}$$

$$mid\ energy\ peak\ position = \frac{1}{2\pi c}\sqrt{\frac{4(\alpha + 2\beta)}{3m_{Si}}}$$

$$high\ energy\ peak\ position = \frac{1}{2\pi c}\sqrt{\frac{2(\alpha\sin^2(\theta/2) + \beta\cos^2(\theta/2))}{m_O}}$$

where $m_O$ and $m_{Si}$ are the atomic masses of oxygen and silicon, respectively.

For glasses, a bond angle between 120° and 180° may be expected with a most likely value of 144° [58,59]. Bond angles are 144° for $\alpha$-quartz and $\alpha$-cristobalite [59,60]. The tridymite group presents a problem as several forms exist including those with 180° angles. Takada et al. [61] showed considerable temperature dependence and noted that a 180° angle may arise from time averaging for tridymite.

The calculation presents a problem as to which peak-position values to use—weighted average, the unperturbed energy prior to LO-TO splitting, TO or LO [15]? Values of less than 140° were seen for silica glass using a $TO_3$ value of approximately 1020 cm$^{-1}$ [57] and the undifferentiated low- (470 cm$^{-1}$) and medium-energy (800 cm$^{-1}$) peaks. We have used the calculated TO band positions (Table 5) to deduce indicative bond angles (Table 6). These figures appear anomalously high even though the $\alpha$ and $\beta$ terms are similar to those found for silica [56,57]. The cause of this is the value of the high-energy peak at approximately 1100 cm$^{-1}$ in all forms of transmission and reflectance spectra. Using the raw transmission mode spectra gave larger angles with a value approaching 180° for opal-CT. An anomalously high figure has been noted before for silica glass [17] and there are several reasons for treating these results with some caution. For instance, the figure for opal-C would be expected to be 144°, rather than 156°, by analogy with cristobalite. Additionally, the data from Table 5 are both from single samples and are Kramers–Kronig manipulated. Small differences in the position of the low- and medium-energy bands are likely to have a large effect on the $\alpha$ and $\beta$ terms. Finally, the applicability of the theoretical calculation has been questioned [15].

**Table 6.** Calculation of bond angles based on TO band positions given in Table 5.

| Type | $\beta$ (Nm$^{-1}$) | $\alpha$ (Nm$^{-1}$) | Si-O-Si Bond Angle (°) |
|---|---|---|---|
| Opal-AG | 99 | 592 | 146 |
| Opal-AN | 101 | 590 | 152 |
| Opal-CT | 103 | 566 | 167 |
| Opal-C | 104 | 570 | 156 |

### 5. Discussion

A coherent picture has emerged relating to peak positions and assignments derived from the transmission and reflectance measurements. The key findings are that both TO and LO components are present for all three major bands, refractive indices are very high for some regions of the infrared spectrum and some peaks show polarisation effects. As known previously, opal-A, opal-CT and opal-C can be readily differentiated by the presence or absence of peaks at 550 cm$^{-1}$ (opal-A) and 620 cm$^{-1}$ (opal-C). The characteristic broad shape of the medium-energy band is a further differentiating feature for opal-A. Classification can be readily achieved using the raw, unprocessed data derived from

transmission (Table 3) or specular reflectance-derived absorptivity (Table 4), even though they represent combined TO/LO modes. There are significant peak position differences for the low- and medium-energy bands in the transmission mode spectra between the different opal types. In general, all observations appear typical for silica and suggest no properties unique to opal.

More information, possibly relevant to structural interpretation, is revealed through Kramers–Kronig analysis of specular reflectance measurements that gives delineated TO and LO peak positions (Table 5) with LO bands at higher energy. All types of opal show a pair of $TO_2/LO_2$ peaks separated by approximately 10 cm$^{-1}$ similar to that seen for cristobalite [38] (785 and 800 cm$^{-1}$, respectively) via polarised specular reflectance. This proximity results in merging to a single peak for all opal types in specular reflection measurements. This is not the cause of the wide peak in opal-A as the individual TO/LO components are broad. The low- and high-energy bands are, however, clearly affected by the intrusion of LO contributions.

$TO_1$ and $LO_1$ are closer for opal-A than for opal-CT and this might explain why the low-energy peak shows a shoulder in the specular reflectance spectra for opal-CT and not opal-A. A $LO_1$ band was not reported in the IR for glass [15]. More influential are the widely separated $TO_3$ and $LO_3$ bands which make the higher-energy region difficult to interpret. This pattern has been widely reported for silica (Table 2) with $LO_3$ bands above 1260 cm$^{-1}$. Visually, the growing $LO_3$ peak in reflectance spectra appears to shift to higher energy with incidence angle. This behaviour is also predicted using the Fresnel equations for reflectance and the Kramers–Kronig-derived absorptivity and refractive indices from Figure 6.

Table 7 provides a comparison of the data for opal-A and opal-CT discovered in this work. Specific TO and LO bands (from the Kramers–Kronig transform of specular reflectance data) are compared with unprocessed spectra from other techniques. Given the variability seen between samples and the extensive use of correction algorithms we mostly give indicative figures (to 5 or 10 cm$^{-1}$ resolution). It is of note that different techniques emphasize different aspects of the spectrum. For instance, in flat sample ATR the medium-energy band is relatively enhanced while the diffuse reflectance spectra show a different structure to those from specular reflectance. Curve fitting of the diffuse reflectance gives a concordant $TO_3$ peak to that from Kramers–Kronig manipulation though the second peak is displaced from that for $LO_3$ by this method. Silanol peaks (see later) are also enhanced.

Finally, we note the empirical observation that the TO bands, particularly $TO_3$, appear to be suppressed with *s*-polarisation for oblique angle specular reflectance, thus increasing the prominence of the LO peak. While $LO_1$ and $TO_1$ may be affected similarly for opal-CT, the smaller splitting makes analysis difficult.

The data provided here may assist in determining constituent silica structure, though it must be stressed that we are unable to identify any candidate structures at this stage. Rather, proposed structures would need to be modelled to determine whether they are consistent with the peak positions and polarisation effects across the various measurement techniques. The literature contains many examples of geometries [23,24,34] that may be tested in such a way as has been done for crystalline forms of silica such as cristobalite [36] and tridymite [61,62]. However, this is not a simple task as, for example, opal-AG may comprise several different Si centres and any structure may be complicated by the presence of water or silanol. For instance, a previous study showed ten discrete layers [63] may be present. The observation of LO bands and polarisation effects may, however, prove valuable in determining symmetry.

The very high refractive indices in the infrared region of the spectrum, in some regions exceeding that for diamond, cast doubt on the appropriateness of using ATR as an analytical tool for opal. This is not unexpected, given the high values have also been reported for silica glass [54] coincident with the low- and high-energy peaks. Our belief is that the conditions for conventional ATR are not followed since the refractive index is higher than diamond in the low- and higher-energy regions. This is likely to be the cause of the variable

behaviour that prompted initial interest in this study. Thus, we believe the major factor is spectrometric rather than compositional. This also has implications for in-built ATR corrections in instrument software where attempts are made to correct for refractive index changes. We suggest only using a simple depth correction to compare spectra with the knowledge that peak positions will be shifted from those observed in transmission because of the anomalous dispersion.

**Table 7.** Compiled spectroscopic data for opal A and opal CT samples. Values given are approximate.

| Feature | Technique | Opal-AG and Opal-AN | Opal-CT |
|---|---|---|---|
| Low-energy bands | Transmission mode ($TO_1$) | Peak approximately 469 $cm^{-1}$. | Peak at 472 $cm^{-1}$ with variable shoulder at 510 $cm^{-1}$. |
| | Specular reflectance (absorptivity index) | Peak approximately 468 $cm^{-1}$. | Peak at 474 $cm^{-1}$. |
| | Specular reflectance ($TO_1$) | Peak approximately 460 $cm^{-1}$. | Peak at 470 $cm^{-1}$. |
| | Specular reflectance ($LO_1$) | Peak approximately 504 $cm^{-1}$. Only appears with KK [a] transformation. | Peak at 536 $cm^{-1}$. Appears in 75° incidence specular reflectance. Enhanced with *p*-polarisation. |
| | Powder ATR | Peak coincident with high refractive index. Ranges from a relatively sharp peak at 470 $cm^{-1}$ to a broad peak at 450 $cm^{-1}$ with a long tail to low energy. | Peak coincident with high refractive index. Ranges from a relatively sharp peak at 475 $cm^{-1}$ to a broad peak at 460 $cm^{-1}$ with a long tail to low energy. |
| | Flat sample ATR | Peak coincident with high refractive index. At approximately 410 $cm^{-1}$ with a long tail to low energy. | Peak coincident with high refractive index. At approximately 410 $cm^{-1}$ with a long tail to low energy. |
| Medium-energy bands | Transmission mode ($TO_2$) | At 802 $cm^{-1}$. Broader than for opal-CT. | At 793 $cm^{-1}$. Narrower than for opal-A. |
| | Specular reflectance (absorptivity index) | Peak approximately 803 $cm^{-1}$. | Peak approximately 792 $cm^{-1}$. |
| | Specular reflectance ($TO_2$) | Peak approximately 800 $cm^{-1}$. Broader than for opal-CT. | Peak at 790 $cm^{-1}$. Narrower than for opal-A. |
| | Specular reflectance ($LO_2$) | Peak approximately 808 $cm^{-1}$. Only resolved by KK transformation. | Peak at 798 $cm^{-1}$. Only resolved by KK transformation. |
| | Diffuse reflectance | At approximately 799 $cm^{-1}$. Broader than for opal-CT. | At approximately 789 $cm^{-1}$. Narrower than for opal-A. |
| | Powder ATR | In the range 790–800 $cm^{-1}$. More prominent than for specular reflectance. | In the range 780–790 $cm^{-1}$. More prominent than for specular reflectance. |
| | Flat sample ATR | At 780 $cm^{-1}$ and more prominent than for powder ATR. | At 780 $cm^{-1}$ and more prominent than for powder ATR. |
| Silanol bands | Transmission mode | Shoulder at approximately 945 $cm^{-1}$ (opal-AG) and 950 $cm^{-1}$ (opal-AN). Possibly coincident with Raman bands. | Not apparent. |
| | Diffuse reflectance | At approximately 945 $cm^{-1}$ (opal-AG) and 950 $cm^{-1}$ (opal-AN). Possibly coincident with Raman bands. | At approximately 970 $cm^{-1}$. Possibly coincident with Raman bands. |

**Table 7.** *Cont.*

| Feature | Technique | Opal-AG and Opal-AN | Opal-CT |
|---|---|---|---|
| High-energy bands | Transmission mode (TO$_3$) | Approximately 1100 cm$^{-1}$ with severe baseline asymmetry. | Approximately 1100 cm$^{-1}$ with severe baseline asymmetry. |
| | Specular reflectance (absorptivity) | Peak approximately 1100 cm$^{-1}$. | Peak approximately 1100 cm$^{-1}$. |
| | Specular reflectance (TO$_3$) | Peak approximately 1085 cm$^{-1}$. | Peak at 1090 cm$^{-1}$. |
| | Specular reflectance (LO$_3$) | Peak approximately 1255 cm$^{-1}$. Enhanced relative to TO$_3$ with *p*-polarisation. | Peak at 1254 cm$^{-1}$. Enhanced relative to TO$_3$ with *p*-polarisation. |
| | Diffuse reflectance | Peak approximately 1090 cm$^{-1}$. Broad. | Peat at 1100 cm$^{-1}$. Broad. |
| | Powder ATR | Peak coincident with high refractive index. In the range 1060–1100 cm$^{-1}$. | Peak coincident with high refractive index. In the range 1065–1100 cm$^{-1}$. |
| | Flat sample ATR | Peak coincident with high refractive index. In the range 1000–1010 cm$^{-1}$ | Peak coincident with high refractive index. In the range 1000–1030 cm$^{-1}$. |

[a] Kramers Kronig transform.

None of the spectra, including transmission or specular reflectance, give any indication of mixed silica species with relatively sharp peaks (e.g., the transmission spectra). For instance, progressive grinding of opal-AG might expose different compositions of silica species as the spheres were destroyed. SEM also supports this as no obvious degradation of the spherical structure was observed. Heterogeneity may also occur through other means. For instance, cationic motion [20] is said to be associated with the 470 cm$^{-1}$ peak though we have no reason to believe that any major differences in structure as the XRD patterns were homogenous. Counterion analysis has been used extensively as a means to establish provenance (e.g., [64]) and diagenesis (e.g., [65]) though localisation is less examined but our belief is that differences would have been seen in broadness or multiplicity of the transmission mode peaks, particularly at 470 cm$^{-1}$. Mineral impurities such as clays or other minerals are again unlikely to be a factor as they are not seen in the XRD patterns nor in the SEM images. Drying experiments revealed no major change other than the usual measurement-to-measurement variability. It is also noted that some of the measurements are performed under vacuum and some not according to the dictates of the instrument protocol, without obvious changes in spectrum. For example, variable incident specular reflectance and the use of the integrating sphere gave consistent spectra. The consistency between the many types of measurement also suggest no surface modification effects. Similarity of spectra after several months ambient storage also mitigate against any crystallization effects.

Variable particle size remains a viable option though we believe not for the usual observation of increasingly sharpened peaks with grinding [66] as the effect is contrary in this study and in any case results in significant peak shifts. Such variability is not a new phenomenon for silica though as far we are aware this has been little explored for opal. As an example, the higher-energy peak for powdered quartz reversibly shifts from 1090 to 1060 cm$^{-1}$ upon wetting [47,67]. Differing sizes of soda lime glass spheres also have very different IR spectra caused by changes in packing density [47]. We propose that each measurement is subject to unique experimental and sample conditions and this causes the measurement-to-measurement variation seen in powder ATR. The correlation shown in Figure 11 supports this notion with the simultaneous effect on both the peak shape and

position. The observation of polarisation effects (Figure 12) also implies non-standard ATR conditions with reflectance-like properties seen.

The "flat sample" ATR spectra further support this explanation. Even very minor changes in position cause very different loadings of the $LO_3$ bands (Figure 13) possibly because of variable contact with the diamond plate. Similar red shifts to lower energy have been reported for glasses [55] and ascribed to refractive index effects.

While these comments cast doubt on the interpretation of the ATR measurements, the technique is still valuable. All forms of opal are readily distinguished using the criteria listed above, including the form of the medium-energy peak. This applies both to the powder and "flat sample" modes. There are also two possible applications. While we are unable to explain the shift in position for the low- and higher-energy peaks for "flat sample" ATR, a fortuitous result is that the characteristic opal-AG and opal-AN peak at 550 cm$^{-1}$ becomes well delineated as the low-energy peaks shifts towards 400 cm$^{-1}$. In theory, this allows delineation of POC opal-AG from POC opal-CT samples (see Figure 13). However, as Figure 14a shows the situation is not clear cut. While the Australian (opal-AG) gems show the peak at 550 cm$^{-1}$ there is still, though of lesser intensity, a peak present for an Ethiopian-sourced cabuchon. Raman spectroscopy confirms this as opal-CT. The second point is more cautionary. We have found that both powder-like ATR and "flat sample"-like spectra may be obtained depending on specific sampling conditions when placed on the ATR plate (e.g., Figure 14b). In another example, G9260 a "milk" opal-AG from an unknown location in Australia displayed both types of spectra by merely altering the position slightly on the diamond plate. Similarly, G9183 (opal-CT from a quarry between Cowell and Mangallo in South Australia) and G32752 (opal-CT from Afar, Ethiopia) showed the "flat sample" spectrum from a cut surface and one like the lightly ground samples on an uncut face. Potentially misleading information may thus be obtained without adequate sample knowledge or preparation method.

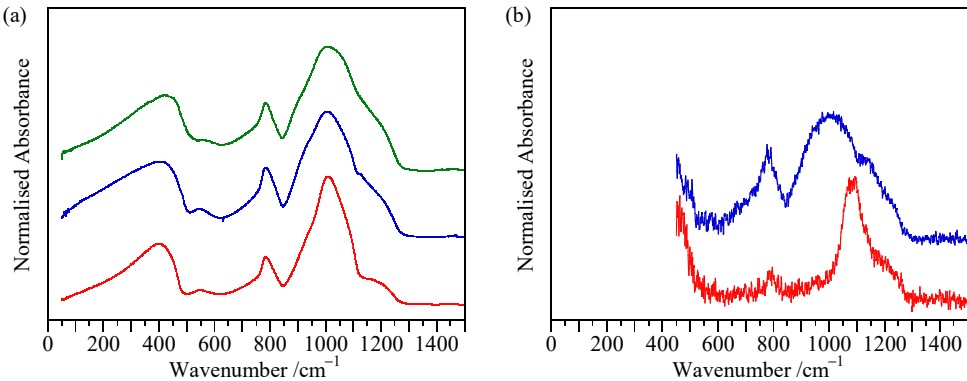

**Figure 14.** Further examples of flat sample ATR spectra: (**a**) opal-AG gems recorded on Bruker v80 instrument: (red) facetted Australian jelly opal, (blue) polished Australian jelly opal, and (green) cabuchon from Ethiopia. (**b**) Untreated opal-CT recorded on Nicolet Nexus 8700: (red) G32226 from St Austell, England, and (blue) G32925 from Tselinograd, Qazaqstan. Spectra are offset and scaled for convenience and corrected for ATR depth penetration.

There is a growing body of evidence to suggest that the macroscopic differences (Figure 1) between opal-AG and opal-AN may be due to geometry imposed by the silanol ($Q_3$) centres [7]. The enhanced peaks at approximately 950 cm$^{-1}$ in the diffuse reflectance spectra are different for opal-AG and opal-AN (Figure 9b). The origin of these peaks is consistent (Table 2) with that for silanol [16,21] though it has been reported that they occur at slightly different positions in the IR and Raman spectra [41]. $^{29}$Si NMR also shows differences in the $Q_3$ peak position (but not $Q_4$) between the two types of opal A [7]. Both transmission mode and powder ATR spectra show a shoulder in this position for opal-AG and opal-AN while they are weak in specular reflectance. The shoulder is, however, not

sufficiently resolved to delineate the two types. Kubelka–Munk-corrected peaks at 937 (opal-AG) and 948 cm$^{-1}$ (opal-AN) indicate a substantial difference in environment.

Given the favourable delineation of the presumed silanol bands in the diffuse reflectance spectra we conducted further experiments to examine the nature of "opaline silicas" [1] such as geyserite/sinters [68] and "Spanish" menilites. These show characteristic opal-A XRD patterns but have different appearance under the SEM to opal-AG and opal-AN. Both show irregular structures including zones of large spheres (Figure 15). This has been seen before for geyserites [69,70]. The "Spanish" menilites have near-white botryoidal form and are distinct from a museum sample described as from Ménilmontant, France (liver opal) which is opal-CT. To this list may be added G34475 described as a hydrophane opal from Dubnik, Slovakia. SEM showed two distinct phases (Figure 16) though both exhibited the characteristic opal-A XRD pattern. For completeness we show Gilson synthetic opal that shows uniformly stacked, but not visibly bonded, spheres (Figure 15).

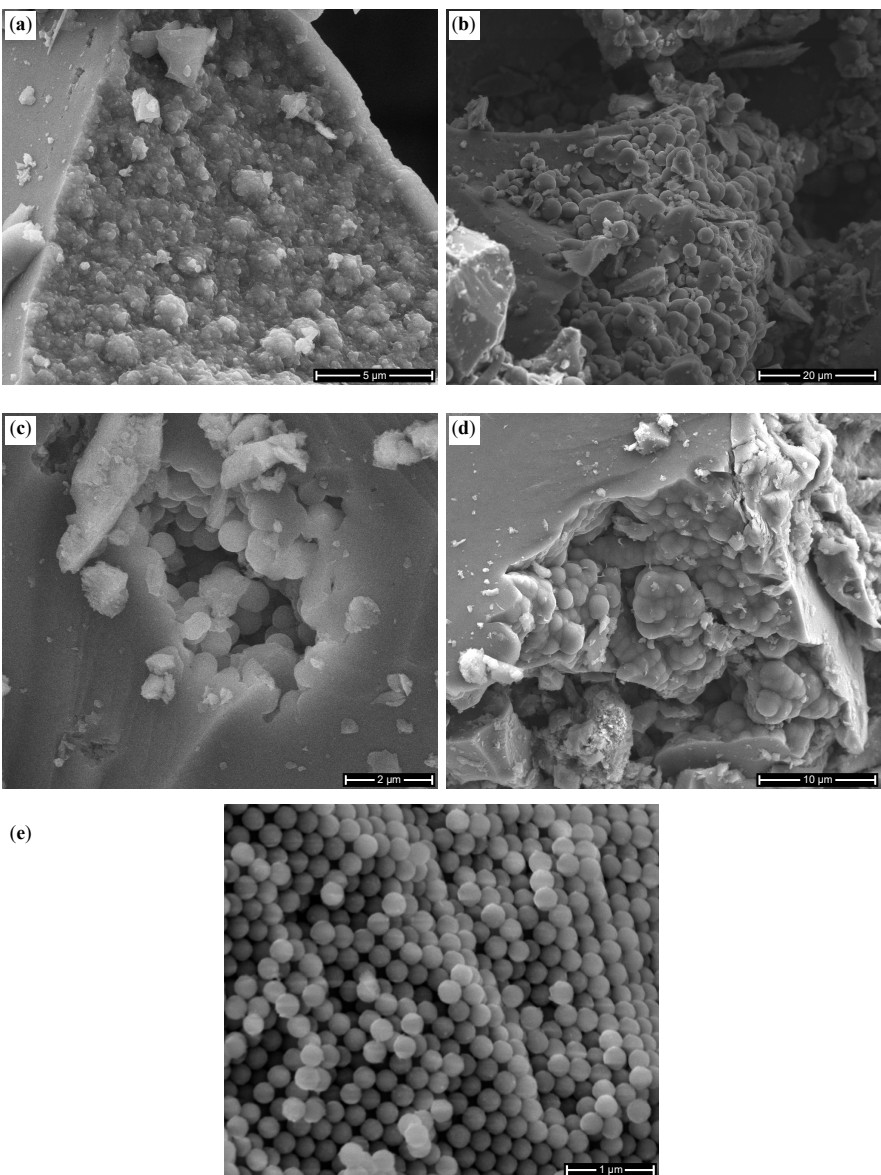

**Figure 15.** SEM images of selected zones of "opaline" silica samples and Gilson synthetic opal. (**a**) G21471 (geyserite from Rotorua, New Zealand), (**b**) T1665 (geyserite from Rotorua, New Zealand), (**c**) G NEW23 (Spanish menilite from Caldes de Malvella, La Selva, Catalonia, Spain), (**d**) G NEW01 (Spanish menilite from Caldes de Malvella, La Selva, Catalonia, Spain) and (**e**) Gilson synthetic opal. Note differences in scale.

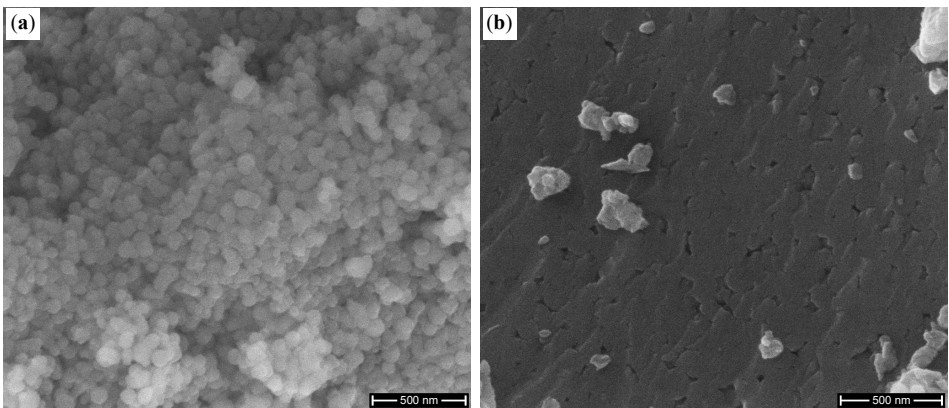

**Figure 16.** SEM images showing two distinct zones for G34475 (opal-A from Dubnik, Slovakia): (**a**) dull opaque zone and (**b**) white translucent zone.

The diffuse reflectance spectra shown in Figure 17a (samples of geyserite and "Spanish" menilite) did not readily fall into either of the sample groups. While the maximum for the presumed silanol band in geyserite is similar to those for opal-AG, the minimum between this and the higher-energy bands is deeper and this may affect the actual position. This difference, of course, may simply relate to characteristic variability of IR measurements though diffuse reflectance is probably, along with transmission, the least likely to be so affected. The Spanish menilite shows a weaker peak, suggesting less silanol, which probably has a maximum less than 950 cm$^{-1}$. The peak at 880 cm$^{-1}$ may be related to an impurity noted at 2.88 Å in the XRD pattern for this and another sample from the same site (see Appendix B).

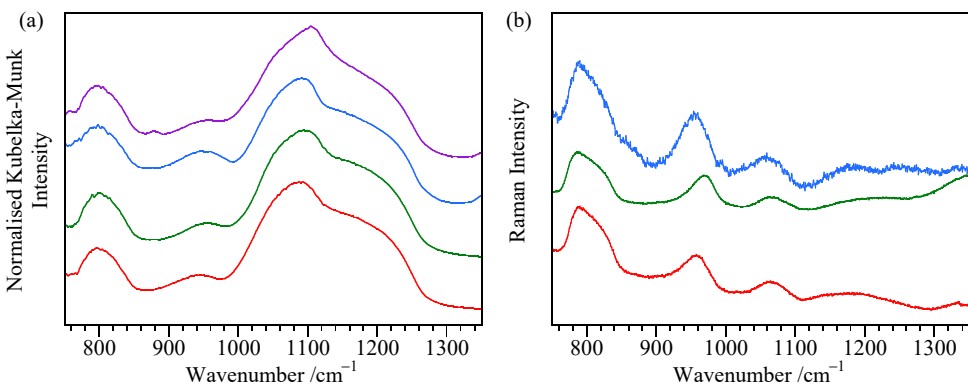

**Figure 17.** Comparison of presumed silanol region for different forms of opal-A: (**a**) diffuse IR reflectance spectra of samples suspended in 5 wt% suspensions in KBr: (red) opal-AG (G9594 from Coober Pedy, South Australia, Australia), (green) opal-AN (MS-4 from Mount Squaretop, Queensland, Australia), (blue) geyserite (G21471 from Rotorua, New Zealand) and (violet) Spanish menilite (GNEW 23 from Caldes de Malvella, Catalonia, Spain). (**b**) Raman spectra: (red) opal-AG (G1442 from William Creek, South Australia, Australia), (green) opal-AN (G1419 from Mount Cora, New South Wales, Australia) and (blue) geyserite (T1665 from Rotorua, New Zealand—linear baseline corrected). Spectra are scaled and offset for convenience.

Peaks in the diffuse reflectance spectra are coincident with minor peaks seen in the Raman spectra [7] and we presume they are derived from the same source. They are also better separated in the Raman and we have thus re-examined the Raman spectra of these examples of opal-A using longer scan times and a more sensitive detector attached to the instrument to those recorded previously [7] (Figure 17b). The characteristic broad medium-energy bands (790 cm$^{-1}$) are again present for opal-AG and opal-AN while the silanol

bands are well separated (958 and 968 cm$^{-1}$, respectively). The peaks at 1064 cm$^{-1}$ are consistent with the IR TO$_3$ band. We find that the peak position for geyserite is more like those for opal-AG than opal-AN. Unfortunately, fluorescence precludes Raman analysis of the "Spanish" menilite. We note in passing that these silanol bands exhibit polarised Raman emission.

We also examined the Raman spectra of the two zones of G34475 (described above). The differences in IR and Raman spectra are slight but probably significant. For instance, the peak of the presumed silanol band in the Raman spectrum (Figure 18b) occurs at a slightly lower-energy peak position for the dull opaque zone.

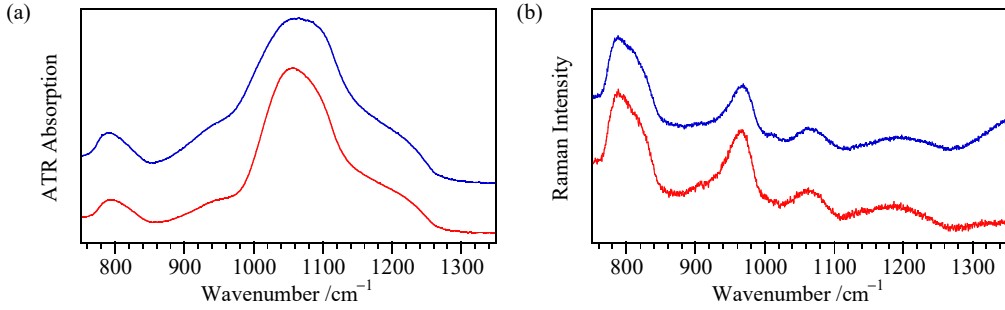

**Figure 18.** Comparison of (**a**) powder ATR IR and (**b**) Raman spectra for a hydrophane opal from Dubnik, Slovakia (G34475) showing minor differences in the presumed silanol region between dull opaque (red) and translucent (blue) zones. Spectra are scaled and offset for convenience.

The implication is that the basic "opal-A" unit can assemble in several different macrostructures, with a characteristic IR and Raman silanol peak. The silicate groups, as shown by XRD patterns and the main Raman and infrared bands, however, are relatively consistent. Thus, another source of potential spectral differences is revealed.

We provide further support for a change in structure in going from those samples of opal-CT with simple XRD patterns (including those with POC) to those with more structure (and more peak delineation in the Raman) [7]. The transmission mode TO$_1$ band (Figure 4) shows a shoulder at 510 cm$^{-1}$ that becomes more apparent (but never major) with the samples with more structured XRD patterns. This may relate to the LO$_1$ band though it is not possible to determine whether it becomes more prominent because of changing species or if conditions become more favourable for its observation. More compelling, Figure 19 shows the appearance of the higher-energy bands in the high angle of incidence specular reflectance spectra. The samples with the more structured XRD pattern show a more pronounced shoulder or peak at 1180 cm$^{-1}$. The evolution may be consistent with the proposed sequence of opal-AG converting to quartz via opal-CT and opal-C [71].

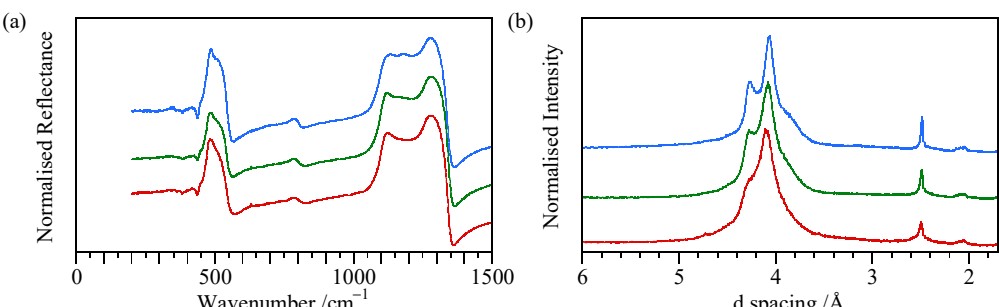

**Figure 19.** Comparison of polished thin opal-CT samples: red) G33912 from Acari, Caravelli Province, Arequipa Department Peru, green) T2222 from Mount Barker, South Australia, Australia, and blue) G9942 from Angaston, South Australia, Australia. (**a**) 75° angle of incidence specular reflectance spectra and (**b**) XRD patterns. Spectra and XRD patterns have been normalised to the tallest peak and then offset for convenience.

A more fundamental change occurs in the Raman spectrum for opal-CT samples between 800 and 1000 cm$^{-1}$. Figure 20 shows powder ATR and Raman spectra of a selection of opal CT samples that are ordered based on their XRD pattern complexity (see Figure 5 in [7]). Only with the more structured samples of opal-CT does the presumed silanol band (Figure 20b) become similar to that seen for opal-A at approximately 960 cm$^{-1}$ [7]. Instead, two samples from Ethiopia show a broad distinct pattern below 950 cm$^{-1}$ but with no obvious peak above 950 cm$^{-1}$. This pattern was also seen in the Raman spectrum of the Ethiopian gem used in Figure 14a. More structured opal-CT samples show residual evidence of this absorption but also the presumed silanol band at ~960 cm$^{-1}$ with slightly varying maxima and intensity. This trend is not apparent in the IR spectra of the equivalent region for (finely ground) powder ATR, though the inherent variability may obscure the effect (Figure 20a). The presence of the broad 900–940 cm$^{-1}$ peak in the Raman spectra observed for "simple" XRD patterns (see Figure 5 in [7]) does not correlate with of play-of-colour since we have seen opal-CT samples with strong POC having a "complex" XRD pattern and narrow 960 cm$^{-1}$ Raman peak. It is not possible to determine whether the peaks below 950 cm$^{-1}$ are due to silanol but the clear differences suggest that the trend in samples with increasingly structured XRD pattern may be significant. The structural implications are only likely to be resolved with more extensive study possibly using the other IR techniques described in this paper, though Raman spectroscopy may be the method of choice owing to the lack of clutter in the key region.

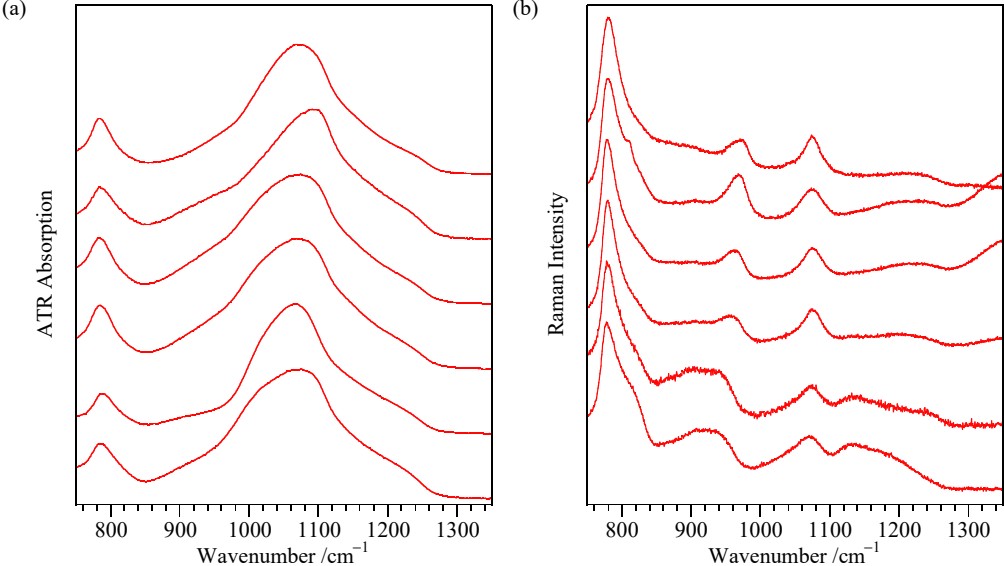

**Figure 20.** Comparison of (**a**) powder ATR IR and (**b**) Raman spectra for a range of opal-CT samples having different XRD patterns. The sample with the least structured XRD pattern is at the bottom and each sample in ascending order have incrementally more complex XRD patterns. Samples in ascending order are: G32752 (from Afar, Ethiopia), NMNH Eth S1 (from Mezezo, Ethiopia), G9964 (from Murwillumbah, Australia), G1441 (from Honduras), M53407 (from Kazaqstan), and G NEW19 (from Tanzania).

Finally, this work can be used to support the notion of transitional opal species [7]. The opal-A to opal-CT transitional samples had a sharpened major XRD reflection and a small but discernable reflection at 2.5 Å. OOC4 and T22824 both have transmission mode low-energy peaks at 471 cm$^{-1}$ more consistent with the figures for opal-CT. These peaks are also relatively narrow and consistent with opal-CT. The medium-energy peaks, however, are broad at approximately 800 cm$^{-1}$ and thus more like opal-A. Both show a weak shoulder at 550 cm$^{-1}$. G14581 (which has a significant quartz reflection in the XRD) also showed the characteristic opal-A shoulder at 550 cm$^{-1}$ in the ATR spectrum (see Figure A2). Specular reflectance gave a characteristic opal-A spectrum with no evidence for a third peak at

$1190 \ cm^{-1}$ at high reflection angles. G NEW18, which had characteristics of (simple [7]) opal-CT and opal-C, had the following characteristics: the narrow $TO_1$ peak is at $472 \ cm^{-1}$ consistent with opal-CT, while the medium-energy peak at $797 \ cm^{-1}$ is more like those found for opal-C. The small peak at $620 \ cm^{-1}$ is also visible in the transmission spectrum.

The observation of high refractive indices, affecting ATR measurements, suggests that transmission mode IR spectroscopy (CsI disc) would be the appropriate method to classify opals if destructive analysis is allowed. Enough data are obtained from the $400–900 \ cm^{-1}$ region for classification and thus the baseline issues for the higher-energy band are not relevant, nor is any further data required, such as from far-IR measurements. However, the current chemistry-practitioner's method of choice is powder ATR owing to ease of use and availability of instrumentation. If this method is used, we recommend using finely ground (to clumping consistency) material and measuring each sample in duplicate as variability will occur. Authentic references are recommended. If non-destructive testing is required or if the sample presents a large and flat area, then other options are available, for instance specular reflectance. "Flat sample" ATR is another option provided that good contact can be made with the diamond plate, though often the signals are weak. It is important to note that the family of spectra from this mode are distinct from the other techniques.

We finish with a consideration of impurity. Opal is not pure silica. Any particular example may have water [72] (free or hydrogen bonded to silanol [73]), silanol [74], hetero-cations (e.g., $Al^{3+}$ [75]), quartz and possibly moganite [76] and still be considered to be opal. The level of free water may be relatively low [77]. There are a mix of silica types within the family of amorphous or poorly crystalline forms of silica that are classified as opal. It is important to address impurities as a possible cause of variation and for differences in the spectra. IR can be used for impure samples, though if foreign material is suspected then confirmation with XRD would be required. Any practitioner's reference library should thus contain data on potential sample-to-sample variability to avoid erroneous conclusions concerning composition. We show four examples that presented indications of minor "impurity" by XRD (Appendix B) and deduce that while minor inclusions of quartz will not interfere with the analysis, there still exists the potential for misleading results for samples studied in isolation.

**Author Contributions:** Conceptualization, N.J.C. and A.P.; methodology, N.J.C. and J.R.G.; software, J.R.G.; validation, N.J.C., J.R.G. and A.P.; formal analysis, N.J.C., J.R.G. and A.P.; investigation, N.J.C. and J.R.G.; resources, A.P.; data curation, N.J.C.; writing—original draft preparation, N.J.C.; writing—review and editing, J.R.G. and A.P.; visualization, N.J.C. and A.P.; supervision, A.P.; project administration, A.P.; funding acquisition, A.P. All authors have read and agreed to the published version of the manuscript.

**Funding:** This research received no external funding.

**Institutional Review Board Statement:** Not applicable.

**Informed Consent Statement:** Not applicable.

**Data Availability Statement:** The data presented in this study are available on request from the corresponding authors.

**Acknowledgments:** The authors again thank the collection managers of the South Australian Museum, the Tate collection at the University of Adelaide, Flinders University of South Australia, Museum Victoria and the Smithsonian National Museum of Natural History. The authors acknowledge the facilities, and the scientific and technical assistance, of Microscopy Australia and the Australian National Fabrication Facility (ANFF) under the National Collaborative Research Infrastructure Strategy, at the South Australian Regional Facility, Flinders Microscopy and Microanalysis, Flinders University. We acknowledge access to the facilities at the Australian Synchrotron and the technical support provided by Ruth Plathe for the Far-IR data collection. We thank Ula Alexander (Flinders University) for assistance in collecting the IR transmission mode spectra at the Australian Synchrotron. The Bruker Vertex v80 FTIR instrument used extensively in this study was purchased using the Australian Research Council (ARC) Linkage Infrastructure, Equipment and Facilities (LIEF) grant LE190100161.

**Conflicts of Interest:** The authors declare no conflict of interest.

## Appendix A　Additional Flat Sample ATR Spectra

This material is presented here as it was obtained from the Nicolet Smart Orbit instrument with a cracked ATR diamond stage; repeatability is thus compromised. Significant variation in the intensity of peaks corresponding to the presumed $LO_3$ bands is seen. This illustrates the potential pitfalls of the technique if the diamond plate is not completely flat.

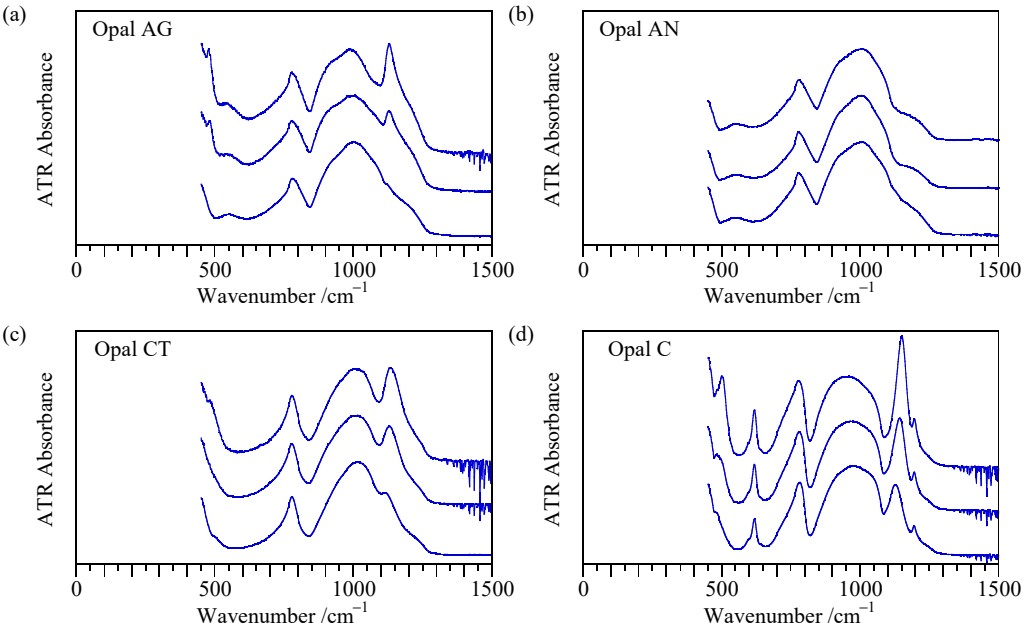

**Figure A1.** ATR of 3 different positions of resin embedded and polished slabs of (**a**) G7532 (boulder opal (AG) from Queensland) (**b**) MS-1 (opal-AN from Mount Squaretop, Queensland, Australia (**c**) G9942 (opal-CT from Angaston, South Australia, Australia, and (**d**) T1664 (opal-C from Guanajuanto, Mexico). Samples were recorded on the Nicolet instrument with Smart Orbit ATR accessory having a cracked diamond crystal.

## Appendix B　Study of Impurities

Figure A2 shows data for powder spectra for G15584 "opal replacing wood" from White Cliffs, New South Wales, Australia. SEM shows regular zones akin to opal-AG as well as loosely ordered spheres depending on zone observed. The level of quartz by XRD is very low and probably much less than 1%. The powder ATR is more complex than the other opal-A samples with a peak at 690 cm$^{-1}$ consistent with for quartz [35]. Again, the trend is towards a lower-energy major peak with grinding.

Three other examples of minor impurities (by XRD) are shown in Figure A3. G14581 (transitioning [7] POC opal-A to opal-CT from Andamooka, Australia) while showing a significant reflection for quartz (at 3.33Å) shows no evidence for it in the IR spectrum. In contrast (though not shown here), G14814 (opal-AG from Piau, Brazil) shows a much larger quartz reflection in the XRD pattern as well as several extra peaks in the 600–800 cm$^{-1}$ in the mid-IR consistent with the known IR bands. G NEW04 (described as green prase from Mt Lyobo Tanzania) which shows a small extra XRD reflection at 4.5Å also appeared typical for opal-CT though with a shoulder at 1000 cm$^{-1}$. G13743 (opal-CT from Siebengebirge, Germany) showed only a very minor extra XRD reflection at approximately 2.5 Å and yet had a shoulder in the infrared spectrum approximately 900 cm$^{-1}$ and a peak at 280 cm$^{-1}$.

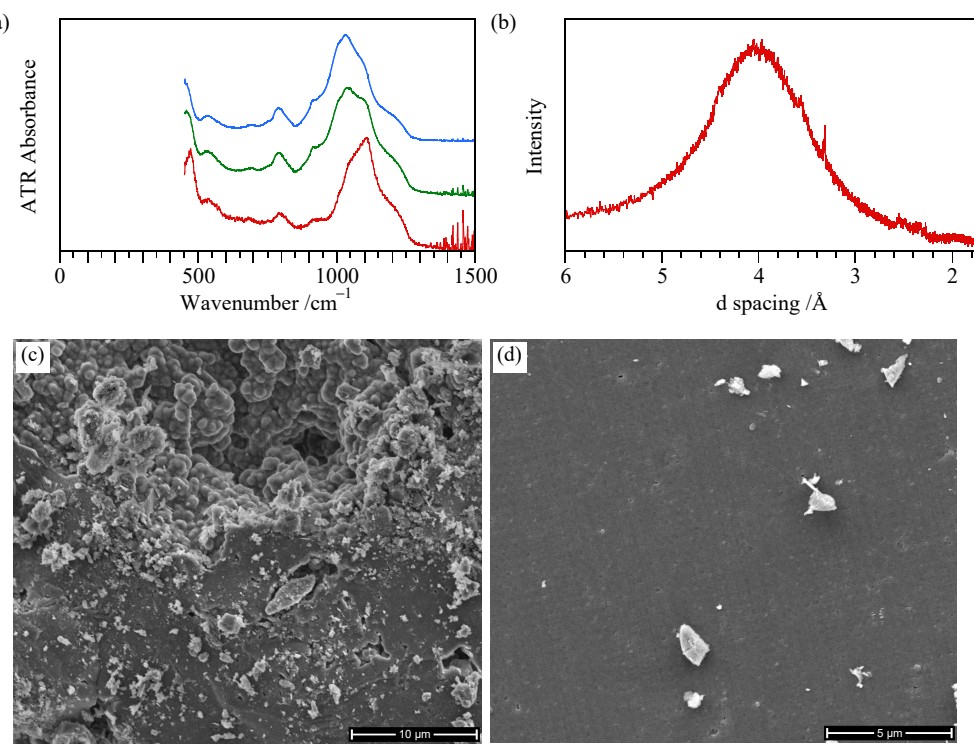

**Figure A2.** Spectra for G15584-AG (opal replacing wood from White Cliffs New South Wales, Australia (**a**) IR ATR spectra of progressively ground samples from the coarsest (red) to finest (blue), (**b**) XRD pattern (Co source), and (**c,d**) SEM images of different regions of a powdered sample. IR spectra are offset and scaled for convenience.

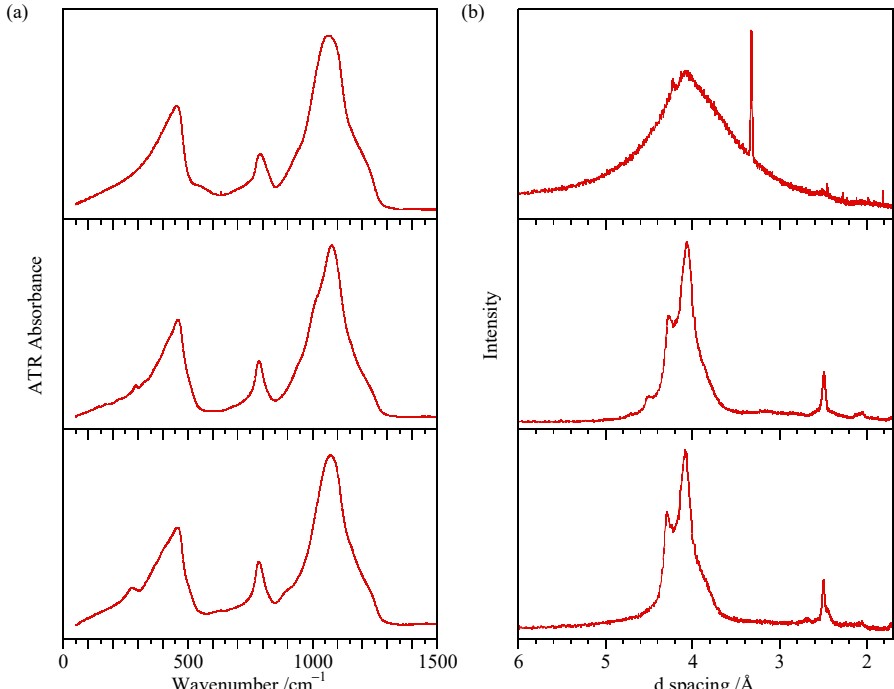

**Figure A3.** Characterisation of samples showing "impurities" in their XRD pattern (**a**) ATR spectra of finely ground samples and (**b**) XRD pattern. In ascending order, G13743 (opal-CT from Siebengebirge, Germany), G NEW04 (opal-CT from Mt Lyobo Tanzania) and G14581 (transitioning opal-A to opal-CT from Andamooka, Australia).

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
