# Peer review of "Silicon-Oxygen Region Infrared and Raman Analysis of Opals: The Effect of Sample Preparation and Measurement Type"

_minerals, doi:10.3390/min11020173_

Round 1

Reviewer 1 Report

Reviewer comments

Minerals-1035164 by Neville J. Curtis and co-authors: Silicon-oxygen region infra-red and Raman analysis of opals: the effect of sample preparation and measurement type  

Dear authors and editor, thank you for the opportunity to review this manuscript about advanced spectroscopic methods used to characterize a variety of opals. With this manuscript the authors provide a well-illustrated extensive dataset for opaline materials, extending their previous study on, for example, opal XRD and Raman spectroscopy characteristics. The text is well-written and refer to the relevant literature. The order and structure of the paragraphs is reasonable and the figures are well-chosen and relevant. In conclusion, this is an excellent piece of work. I am pleased to recommend it for acceptance after minor amendments.

Abstract: L22-23: X-ray powder diffraction.

General remark: Use the terms peak or band consistently. They are not synonymous. For XRD it should be peak. For IR and Raman spectra it should be band. This must be corrected throughout the entire manuscript.

L36 (and, e.g., 58, 236): “POC arises if the lepispheres in opal-AG..”

            The “particles” in opal-AG are spheres. Lepispheres are observed in opal-CT.

L34 (and, for example, L47 and 548): low/α-cristobalite

L39-41: “Etching of opal-AG… is  not  required  to  show  the arrangement,  uniformity  and  size  of  the  spheres.”. 

Figure 1c actually appears to be the only unetched sample showing individual spheres. The others show ordered/disordered pores in a more or less homogeneous mass. Maybe the sentence in L39-41 could be a little adjusted accordingly.

L112: Spectra instead of samples.

L113: Band instead of peak.

L158-159: “XRD patterns were collected as before [7].” There is enough space in the manuscript to include at least one or two sentences about the measurement conditions. This should include the cathode material, step size, etc.

L236-239: What is the detection limit for Na and K, and what do you mean with cementing medium? The silica cementing the spheres?

L260: “..while the those..”. Delete “the”.

L347: “than” instead of “that”.

L628-629: “Mineral impurities such as clays or other minerals are again unlikely to be a factor as they are not seen in the XRD patterns.”

Keep in mind that the XRD detection limit is about 1 vol.%.

L1086: Do you mean 42°2θ? What is the wavelength used here?

Reviewer 2 Report

Please see attachment below

Author Response

Please see attachment below.

Reviewer 3 Report

Curtis et al. provide a detailed set of diffuse reflectance and ATR IR spectra, Raman spectra, and XRD patterns of a variety of natural opals (opal-A, opal-CT, and opal-C) to better describe the spectral characteristics of opals and their structures. This manuscript will be a useful reference to those in the scientific community who study any variety of amorphous or paracrystalline silica using spectral techniques. The methodology and data presented are sound. I have a few minor comments and edits in the attached document. I recommend this manuscript be published after these minor comments and edits are addressed.
